# `Lens`: A Knowledge-Guided Foundation Model for Network Traffic

**Xiaochang Li[1]   Chen Qian[1]   Qineng Wang[2]   Jiangtao Kong[1]   Yuchen Wang[1]**
**Ziyu Yao[3]   Bo Ji[4]   Long Cheng[5]   Gang Zhou[1]   Huajie Shao[1†]**

**[1]William & Mary   [2]Northwestern University   [3]George Mason University**
**[4]Virginia Tech   [5]Clemson University**
`{xli59@wm.edu, hshao@wm.edu}`

**Reviewed on OpenReview:** https://openreview.net/forum?id=cGDwTgnJIR

## Abstract

Network traffic refers to the amount of data being sent and received over the Internet or any system that connects computers. Analyzing network traffic is vital for security and management, yet remains challenging due to the heterogeneity of plain-text packet headers and encrypted payloads. To capture the latent semantics of traffic, recent studies have adopted Transformer-based pretraining techniques to learn network representations from massive traffic data. However, these methods pre-train on data-driven tasks but overlook network knowledge, such as masking partial digits of the indivisible network port numbers for prediction, thereby limiting semantic understanding. In addition, they struggle to extend classification to new classes during fine-tuning due to the distribution shift. Motivated by these limitations, we propose `Lens`, a unified knowledge-guided foundation model for both network traffic classification and generation. In pretraining, we propose a Knowledge-Guided Mask Span Prediction method with textual context for learning knowledge-enriched representations. For extending to new classes in finetuning, we reframe the traffic classification as a closed-ended generation task and introduce context-aware finetuning to adapt the distribution shift. Evaluation results across various benchmark datasets demonstrate that the proposed `Lens` achieves superior performance on both classification and generation tasks. For traffic classification, `Lens` outperforms competitive baselines substantially on 8 out of 12 tasks with an average accuracy of **96.33%** and extends to novel classes with significantly better performance. For traffic generation, `Lens` generates better high-fidelity network traffic for network simulation, gaining up to **30.46%** and **33.3%** better accuracy and F1 in fuzzing tests. Code is available at https://github.com/V-Enzo/Lens-foundation-model-for-network-traffic

## 1 Introduction

In computer networking, network traffic Oliveira et al. (2016) is defined as the flow of data, transmitted in the form of packets between interconnected computers or systems. The network packet consists of plain-text headers with metadata and an encrypted payload holding actual content. Given the critical role of networking, analyzing network traffic is crucial to ensure high network security, offer high-quality network services, and facilitate effective network management.

Over the past decades, many approaches have been developed for network traffic analysis. Early works Taylor et al. (2017); Al-Naami et al. (2016); Panchenko et al. (2016); Wang et al. (2017b) mainly utilize statistical methods, heavily dependent on manually crafted features. To address this issue, some studies have employed deep learning methods Sirinam et al. (2018); Liu et al. (2019); Lotfollahi et al. (2020); Wang et al. (2018) to extract complicated features from raw data. While these approaches have shown impressive

---

[†]Corresponding author.

results in specific tasks, they often require extensive labeled data and struggle with generalization to new tasks. Consequently, recent works Lin et al. (2022); Zhao et al. (2023); Meng et al. (2023); Cui et al. (2025) have developed foundation models employing various pretraining techniques to learn network representations from large-scale, raw traffic data.

While they show promising results on various tasks, they still face two major *limitations.* (i) Current data-driven pretraining methods Lin et al. (2022); Zhao et al. (2023) overlook vital network knowledge, resulting in incomplete semantic understanding. e.g., masking partial digits "400" of an indivisible port number "40023" for prediction. (ii) In finetuning for classification, most encoder-based models He et al. (2020); Guthula et al. (2023) use an additional multi-layer perceptron (MLP) as the classifier, which performs poorly when extending to new classes as they need to be re-trained for accommodation Li & Hoiem (2017). These limitations motivate us to address two fundamental research questions: **RQ1:** How can we integrate network-specific knowledge into pretraining to learn better network representations? **RQ2:** How can we seamlessly extend traffic classification to novel classes while maintaining high performance?

To answer these questions, we propose `Lens`, a unified knowledge-guided foundation model based on an encoder-decoder T5 architecture Raffel et al. (2020); Wang et al. (2021) to learn network representations from massive raw traffic data. The encoder-decoder architecture is chosen because it better captures the global information of the input data, making it well-suited for the header generation task based on its successor payload. In contrast, decoder-only models pretrained on next token prediction struggle with this due to their auto-regressive nature. Further discussions and ablation studies are shown in the Appendix F. More specifically, to address **RQ1**, we propose the Knowledge-Guided Mask Span Prediction (KG-MSP) to intentionally mask network metadata and payload-related information as a whole based on their importance in networking. KG-MSP integrates domain knowledge about network traffic into the pretraining objective, rather than relying on generic self-supervised signals. Specifically, learning representations of port numbers and protocols as indivisible semantic units is critical for application classification, while preserving protocol flags and key fields as complete entities is essential for interpreting packet intentions and protocol behaviors. Plus, we incorporate context information as auxiliary knowledge into model pretraining. For **RQ2**, we reframe the network traffic classification as a closed-ended generation task for adapting distribution shifts. Then, we leverage context-aware finetuning to smoothly extend classification from known to new classes by training only on new-class data with the updated context.

Finally, we assess the performance of `Lens` on 12 network traffic classification tasks and 5 network generation tasks across 6 datasets Draper-Gil et al. (2016); Habibi Lashkari. et al. (2017); Wang et al. (2017a); Van Ede et al. (2020); MontazeriShatoori et al. (2020); Neto et al. (2023). For the traffic classification, evaluation results show that `Lens` outperforms competitive baselines across 8 out of 12 tasks with an average accuracy of 96.31%. Besides, `Lens` extends to classify novel classes well and gains accuracy and F1 advantage up to 31.31% and 42.59%, respectively. Regarding the traffic generation, `Lens` generates network traffic that more closely aligns with real-world data, achieving superior performance in network fuzzing tests with up to 30.46% higher accuracy and a 33.3% higher F1 score.

Our contributions are summarized as follows: 1) We propose `Lens`, a novel knowledge-guided foundation model for network traffic. The proposed model integrates Knowledge-guided Mask Span Prediction with context for learning generalizable network representations in pretraining. 2) We reframe the traffic classification as a closed-ended generation task to adapt the distribution shift, which enables excellent extensibility by simply updating the textual context and lightly finetuning only on new classes. 3) We evaluate `Lens` on both network traffic classification and generation. For classification, `Lens` outperforms competitive baselines on most tasks and extends to novel classes significantly better. For generation, `Lens` generates high-fidelity network traffic that closely mirrors real-world distributions, enhancing the efficacy of subsequent fuzzing tests.

## 2 Related Work

### 2.1 Network Traffic Classification

**Classical Machine Learning Methods.** Earlier works have employed classical machine learning methods for network traffic analysis. For example, Wang et al. Wang et al. (2017b) used the K-Nearest Neighbors (KNN) to identify attacks. CUMUL Panchenko et al. (2016) adopted SVM for network traffic identification, while APPScanner and BIND Taylor et al. (2017); Al-Naami et al. (2016) used statistical features like temporality and packet size to train Random Forests classifiers for identification tasks. Besides, IsAnon Cai et al. (2019) fused Modified Mutual Information and Random Forest (MMIRF) to filter out redundant features. However, these methods require expert knowledge for feature extraction and lack generalization capability.

**Deep Learning Techniques.** Deep learning techniques have been introduced to provide a more automated approach to comprehend network traffic without human-designed features. DF Sirinam et al. (2018), for example, devised Convolutional Neural Networks (CNN) for identifying a novel website fingerprinting attack. In addition, FS-Net Liu et al. (2019); Lin et al. (2021) employed recurrent neural networks (RNN) and its variant LSTM Yao et al. (2019); Wang et al. (2018) to classify network traffic. Recently, a method called DeepPacket Lotfollahi et al. (2020) has been introduced. It combined the stacked autoencoder (SAE) with CNN to identify and extract the important features for traffic classification tasks. However, these approaches rely heavily on large amounts of labeled data and have limited generalization ability.

**Pre-training Approaches.** To improve model generalization ability, recent studies have adopted pre-training techniques to learn representations from large-scale traffic data in an unsupervised manner. For instance, PERT He et al. (2020) and ET-BERT Lin et al. (2022) leveraged the ALBERT Lan et al. (2019) and BERT Devlin et al. (2018) to learn the latent network representations, respectively. Lately, netFound Guthula et al. (2023) and NetMamba Wang et al. (2024) pre-trained hierarchical Transformers and state space models for learning network representations separately. However, these models pre-train on random masking and are only applicable for classification due to their encoder-only structure. At the same time, Decoder-based models like NetGPT Meng et al. (2023), GBC Zhao et al. (2025), and TrafficGPT Qu et al. (2024) are pre-trained on the next token prediction for both traffic classification and generation. Nevertheless, they predict tokens based on causal probabilities, inferring the value of network fields based on partial context. More recently, works like TrafficLLM Cui et al. (2025) fine-tuned LLM pretrained on natural language for traffic classification and generation. However, it suffers from hallucination and requires more labeled data to mitigate the domain gap. Different from prior works, we pre-train a network foundation model with a knowledge-guided task combined with auxiliary context to learn better network representations. Table 1 summarizes the comparison of the proposed `Lens` and other existing methods.

Table 1: Comparison of the proposed `Lens` and existing pretraining methods. "KG Pretrain" denotes knowledge-guided pretraining. "IP Masking" means all IPs are anonymized or removed in both pretraining and finetuning for privacy. "Generation" refers to the support of generation tasks.

| Method | Encoder | Decoder | KG Pretrain | IP Mask | Generation |
|---|---|---|---|---|---|
| PERT He et al. (2020) | ✔ | ✘ | ✘ | ✘ | ✘ |
| ET-BERT Lin et al. (2022) | ✔ | ✘ | ✘ | ✔ | ✘ |
| NetGPT Meng et al. (2023) | ✘ | ✔ | ✘ | ✘ | ✔ |
| YaTC Zhao et al. (2023) | ✔ | ✘ | ✘ | ✔ | ✘ |
| TrafficLLM Cui et al. (2025) | ✘ | ✔ | ✘ | ✔ | ✔ |
| **Lens (Ours)** | ✔ | ✔ | ✔ | ✔ | ✔ |

### 2.2 Network Traffic Generation

**Tool-Based Traffic Generation.** Classical traffic generation methods mainly focus on simulation tools and structure-based solutions. Simulation tools, such as NS-3 Henderson et al. (2008), yans Lacage & Henderson

(2006), and DYNAMO Bühler et al. (2022), are based on varying network topology. Structure-based methods like Iperf Botta et al. (2012), Harpoon Sommers et al. (2004), and Swing Vishwanath & Vahdat (2009) capture network patterns via heuristics. However, these methods require vast domain expertise and might lack versatility. Moreover, tool-based traffic generation methods often result in rigid traffic patterns that may not accurately reflect the dynamic and stochastic nature of real-world network conditions. Thus, they cannot effectively adapt to the evolving behaviors of cyber threats or user demands.

**GAN-Based Traffic Generation.** In addition, some studies have employed Generative Adversarial Networks (GAN) to generate network traffic. Ring et al. Ring et al. (2019) first suggested the use of GAN Goodfellow et al. (2014) for the simulation of flow-level traffic. The following works include NetShare Yin et al. (2022) and DoppelGANger Lin et al. (2020), and others Mozo et al. (2022); Hui et al. (2022). For instance, NetShare generated packet and flow header traces for networking tasks, such as telemetry, anomaly detection, and provisioning. Though GAN-based methods are adaptive, their generated results may be inconsistent with target protocols Yin et al. (2022).

**Pretraining-based Generation.** Recently, researchers employed a pretraining technique for traffic generation. The Decoder-based NetGPT Meng et al. (2023) and TrafficGPT Qu et al. (2024) have been developed to generate key network header fields. However, it is hard to assess their performance as they did not compare results with the state-of-the-art models like Netshare. More recently, NetDiffusion Jiang et al. (2024) applied diffusion models to generate network traffic, while Chu et al. Chu et al. (2024) investigated the feasibility of state space models in network traffic generation. Nevertheless, our models can adapt to both classification and generation. Most recently, TrafficLLM Cui et al. (2025) fine-tuned LLM to generate network packets, but it requires more data and computational resources to achieve good performance. Unlike previous works, our `Lens` leverages context-aware finetuning to generate high-fidelity network traffic available for downstream simulation.

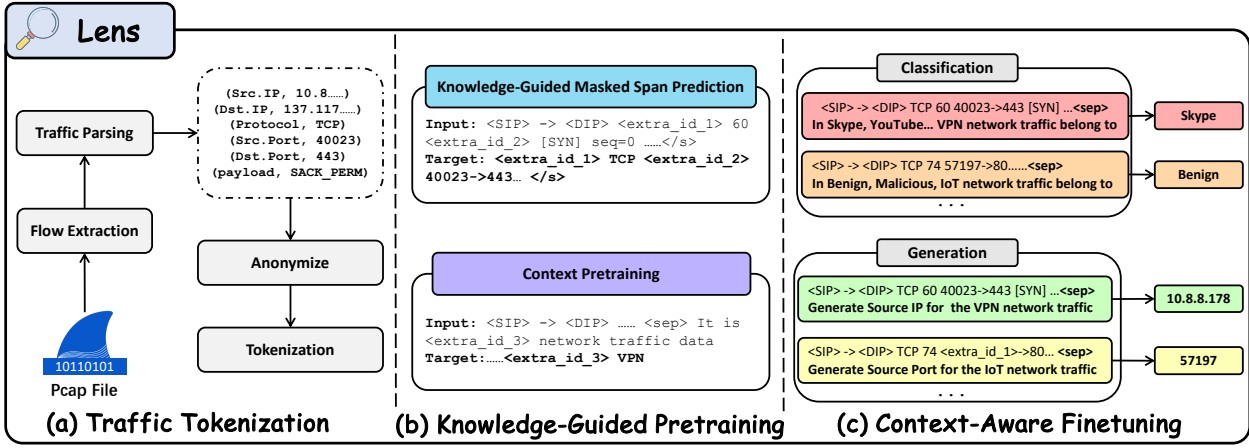

Figure 1: The overall framework of `Lens`. (a) Network flows are extracted, parsed with Tshark, anonymized, and tokenized using our network-specific tokenizer. (b) Lens is pretrained with Knowledge-Guided Masked Span Prediction (KG-MSP) and auxiliary natural-language context. (c) In finetuning, Lens performs downstream classification and generation tasks via context-aware finetuning.

## 3 Overall Framework of LENS

We develop `Lens`, a unified knowledge-guided foundation model for network traffic classification and generation. The Figure 1 illustrates the overall framework, consisting of three main stages: (a) traffic tokenization, (b) knowledge-guided pretraining, and (c) context-aware finetuning. In Section 3.1, we detail how we preprocess and conduct network-specific tokenization to benefit representation learning. Then, Section 3.2 presents the knowledge-guided Masked Span Prediction (KG-MSP) with the context pretraining. Lastly, Section 3.3 demonstrates the context-aware finetuning on downstream network traffic classification and generation tasks.

### 3.1 Traffic Tokenization

To preprocess input data with network knowledge, we parse network flows through Tshark Combs & team (2024) into text-like input as shown in Figure 4 of the Appendix D. Afterwards, we pretrain a specialized Byte-level Byte Pair Encoding (BBPE) Wang et al. (2020) tokenizer on the textual network input to tokenize network terms, like "Seq", "TCP", and natural language context properly. More details about the data preprocessing are elaborated in the Appendix A.1.

### 3.2 Knowledge-guided Pretraining

We pretrain the proposed `Lens` using Knowledge-Guided Masked Span Prediction (KG-MSP), an objective that intentionally masks significant network metadata and payload-related information entirely to learn generalizable representations for downstream tasks. By also incorporating textual context as auxiliary knowledge, `Lens` is able to learn more generalizable representations after pre-training on both network traffic and natural language.

#### 3.2.1 Knowledge-guided Mask Span Prediction

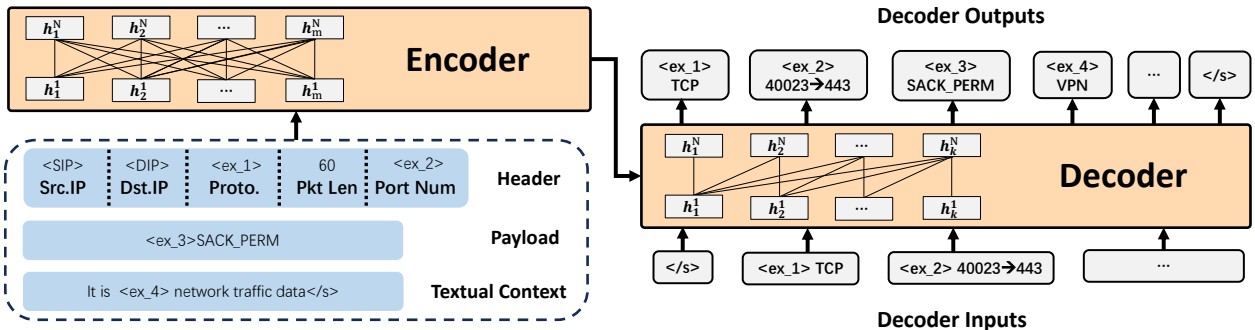

Figure 2: The core model architecture of `Lens` for pre-training with both the encoder and decoder. 1) The encoder takes in masked network traffic (header and payload) and textual template context. 2) The decoder uncovers the masked span tokens in both traffic and context based on Encoder representations in an auto-regressive way.

The data-driven random masking Devlin et al. (2018); Lin et al. (2022) has been proved effective in pretraining across many NLP tasks. The basic idea is to first mask random tokens in the input and then recover them at the decoder side. However, directly applying random masking may degrade network representation, since it may mask network metadata partially. For example, masking partial digits "400" of the port number "40023" for prediction harms the understanding of the port number as a complete semantic unit. Similarly, masking the "T" of "TCP" also hurts the representation of the network protocol.

Motivated by this observation, we propose a new knowledge-guided masking method to intentionally mask vital network metadata such as protocol and port numbers entirely, thereby learning better network representations, as shown in Figure 2. As mentioned before, the parsed network packet contains important network-specific knowledge in both the header and the payload. In both TCP and UDP network traffic, the header meta-information includes anonymized IP addresses, network protocol, packet length, port numbers, and payload length. To ensure network reliability, the TCP packet owns more header fields like sequence, acknowledgment numbers, maximum segment size, and so on. Besides, other text-like network messages like SACK_PERM, Client Hello, indicate the purpose of the sent packet. To learn the generalizable representations of network metadata in pretraining, we set different probabilities to mask them based on their importance in networking.

**For identifying application types, port numbers and protocols are critical.** For example, VPN applications typically use port 443 for secure TCP transmission to protect the content's safety. Besides, most

applications use the DNS protocol for looking up the IP address of a network domain. Given its importance, we set a $\theta$ probability to mask them for learning insightful network representations empirically. Both the source and destination port numbers are masked for a better understanding of the network traffic direction.

**For understanding packet intentions, protocol flags and textual payload messages are essential.** For instance, the network flag "[SYN]" means synchronization of the TCP 3-way handshake, indicating the start of a TCP session. Similarly, the text-like TLS handshake message Server Key Exchange hides the process of exchanging encrypted keys but preserves the process of building connections. Since protocol flags and messages are essential for interpreting underlying interactions within network flows, we also set the same probability of $\theta$ to mask them as a whole. To prioritize learning these important fields' representations, `Lens` assigns higher masking probabilities with $\theta = 60\%$. The sensitivity analysis (Table 15a, Appendix B.3) also confirms the robustness to different $\theta$ values.

**For analyzing protocol behavior, sequence and acknowledgment numbers are fundamental.** They record how many bytes are exchanged between the server and client, ensuring reliable TCP transmission. But, their patterns are comparatively dynamic as the retransmission process or the loss of packets in the network traffic changes them from time to time. Likewise, the packet and payload length vary for more efficient transmission in different network conditions. Thus, we assign a probability of $k$ to mask sequence numbers, acknowledgment numbers, and the length of packets and payloads. We conducted a sensitivity analysis on k (Table 15b, Appendix B.3), which shows robustness to different ratios. We adopt $k = 50\%$ to pretrain more on these significant fields, enhancing `Lens`'s fundamental understanding of protocol behavior.

To preserve the randomness in the masking policy, we further mask randomly on the rest of the input to satisfy the overall 15% masking ratio as Raffel et al. (2020). To calculate the loss function of KG-MSP, we use the following negative log likelihood

$$\mathcal{L}_{\text{KG-MSP}} = -\sum_{i=1}^{k} \log P(\text{KSPAN}_i = \text{SPAN}_i | \mathbf{x}_{\text{in}}, \text{KSPAN}_{<i}; \theta), \tag{1}$$

where $\theta$ represents the model (both encoder and decoder) parameters, $k$ is the total number of masked spans, $\text{KSPAN}_i$ is $i$-th span generated by the decoder part of `Lens`, $\text{SPAN}_i$ denotes the original span tokens, $\mathbf{x}_{\text{in}}$ is the masked input sequence of the encoder, $\text{KSPAN}_{<i}$ indicates the generated spans from the decoder before $i$-th span. $\text{KSPAN}_i$ is generated given $\mathbf{x}_{\text{in}}$ and $\text{KSPAN}_{<i}$ when parameterized by $\theta$. Here, `Lens` decodes the original span tokens after the special token used for masking in input as Raffel et al. (2020).

To demonstrate KG-MSP's superiority, we compare KG-MSP with random masking used in ET-BERT Lin et al. (2022). In essence, it masks overall 15% of the input randomly on single tokens for prediction. We pretrain the random masking with `Lens`'s base model and evaluate on the downstream classification task. The ablation results in Table 12 in the Appendix B.1 demonstrate the superiority of KG-MSP.

**Context Pretraining.** Aside from learning better network representations via KG-MSP, we combine the textual description of network traffic input as auxiliary knowledge. As shown in Figure 2, the textual description like *It is VPN network traffic data* is appended after the network traffic with the special "<sep>" token in the middle separating the two modalities. The textual description is derived solely from the dataset source and does not contain any downstream task labels, ensuring that no task-specific information is leaked. The benefits are two-sided: 1) The auxiliary description helps differentiate similar network traffic via its sources, like VPN, DoHBrw, IoT, and so on. 2) The textual description bridges the gap between network traffic and natural language, preparing basic natural language understanding for downstream context-aware finetuning. To pretrain the auxiliary knowledge, we randomly mask 15% of the textual description as NLP pretraining.

### 3.3 Context-aware Finetuning

After knowledge-guided pretraining, we fine-tune the proposed `Lens` on various downstream network traffic classification and generation tasks. For the traffic classification, `Lens` reframes it as a closed-ended generation task and then leverages context-aware finetuning to generate the label at the decoder side. This reformation enables `Lens` to adapt distribution shifts easily and extend to classify new classes, which benefits generaliza-

tion in fast-changing network environments. In addition, `Lens` can also be fine-tuned to generate high-fidelity network traffic for better downstream network simulation, such as fuzzing tests.

**Network Traffic Classification.** As shown in Figure 1 (c), the input starts with the parsed network traffic and the task context with the format: *network traffic <sep> In Skype, Youtube, . . . , the VPN network traffic belongs to.* In the task context, we provide all application labels for `Lens` to select. This provides potential for extensibility to unseen classes through adding more label options in the context. After feeding the network traffic and the task context to the encoder, `Lens` generates labels on the decoder side based on input representations. To prevent generating labels out of choices, we guarantee the validity of the output by using a prefix trie as GENRE De Cao et al. (2021) to constrain the generation process, ensuring the output sequence always matches a known label.

**Extending Network Classification to Novel Classes.** Reforming traffic classification into a closed-ended generation task, `Lens` alleviates the distribution shifts and seamlessly extends to new classes via context-aware finetuning. Specifically, after encountering unseen data from new classes, `Lens` reloads the saved checkpoint to resume the learned knowledge on known classes first. Then, it adds new label options in the task context and only fine-tunes on unseen data, correlating new labels to unseen data. Through this process, `Lens` reuses learned knowledge for classifying new classes seamlessly through discerning the differences in the task context.

**Network Traffic Generation.** For the generation task, the context-aware finetuning also benefits `Lens` significantly through providing auxiliary knowledge. As shown in Figure 1 (c), the input format for the generation task is like *network traffic <sep> Generate Source IP for the VPN network traffic.* To complete different generation tasks, we substitute the task context with various generation targets, such as port numbers, and mask the corresponding fields in the network traffic input for generation. After generating necessary header fields, we evaluate the fidelity of synthetic network traffic on fuzzing tests Munea et al. (2016) to verify its usefulness. Following NetShare Yin et al. (2022)'s setting, generating encrypted payloads is a lower priority, as most real-world payloads are encrypted and uninformative. Instead, replaying captured payloads or applying standard encryption algorithms provides a more realistic and practical reconstruction of network traffic payload.

## 4 Experiments

Extensive experiments are carried out to evaluate the performance of `Lens` on both network traffic classification and generation tasks across 6 different datasets. First, we elaborate on the implementation details and experimental settings, including datasets, baselines, and evaluation metrics. Then, in Section 4.1 and 4.2, we compare our `Lens` with competitive baselines on 12 classification tasks and demonstrate its extensibility to new classes. Afterwards, in Section 4.3, we showcase `Lens`'s traffic generation capabilities and simulation results in fuzzing tests. Finally, we conduct ablation studies to further investigate the impact of the main components on model performance in Section 4.4.

**Datasets.** In our experiments, we use six publicly available datasets in NetBench Qian et al. (2024), including ISCX-VPN Draper-Gil et al. (2016), ISCX-Tor Habibi Lashkari. et al. (2017), USTC-TFC-2016 Wang et al. (2017a), Cross Platform Van Ede et al. (2020), CIC-DoHBrw-2020 MontazeriShatoori et al. (2020), and the CIC-IoT-2023 Neto et al. (2023). Based on these 6 datasets, we conduct 12 downstream network traffic classification and 5 network generation tasks. In the following, we will detail how we pre-process and organize pretraining and finetuning datasets. The detailed dataset distribution between pretraining and finetuning datasets are listed in Table 10 in Appendix A.2.

**Pretraining Data.** For each dataset (except CrossPlatforms), we randomly sample 60% of flows using a stratified sampling strategy, but only with respect to the coarsest dataset-level categories (e.g., VPN vs non-VPN, benign vs malicious), without using any fine-grained task labels such as application or service types. This ensures that pretraining does not access any downstream classification labels and completely avoids label leakage. The pretraining split is also used to pretrain our network-specific BBPE tokenizer, together with the mini C4 natural language corpus Miranda (2021).

**Finetuning Data:** For finetuning, we randomly select around 10k network flows for finetune training for each dataset except CrossPlatform. Since downstream labels like "YouTube" and "Skype" are not trained in the pretraining stage, fine-tuning requires sufficient data to learn these label strings. For tasks with hundreds of classes, we map textual labels to numeric indices, such as denoting "Youtube" as 0, "Skype" as 1, to keep the context compact and avoid prefix overlap. Moreover, we preserve the imbalanced nature in both training and testing, like real-world situation.

We conduct flow-level classification and packet-level generation. Specifically, `Lens` classifies each flow using its first 34 packets and performs packet-level generation using one packet sampled from each flow for universality. In the classification task, the choice of 34 packets is derived from the dataset statistics in Table 9 of Appendix A.1. In the generation task, we mask the header field that the model is required to predict in both training and testing. Representative data examples are shown in Figure 4 in Appendix D.

**Implementation Details.** We adopt Google's T5-v1.1-base model Raffel et al. (2020), which contains roughly 0.25B parameters, as the foundational architecture for the proposed `Lens`. To process the input length up to 1,500, we develop `Lens` based on the TurboT5's implementation Knowledgator (2023). All experiments are conducted on a GPU server with 4 NVIDIA A6000 48G GPUs, Ubuntu (20.04.6). For pretraining, we set the batch size to 48 and implement gradient accumulation over 6 training steps. The learning rate is set to $5 \times 10^{-4}$ over a total of 130,000 steps with 10% steps as a warm-up. For finetuning, we set the batch size to 32, dropout rate to 0.1, and the learning rate to $5 \times 10^{-5}$, training each downstream task for 20 epochs using the AdamW optimizer Loshchilov & Hutter (2017). More details are detailed in Table 26 of the Appendix E.

**Baselines.** To compare the performance of `Lens`, we compare with two types of baselines: (i) `Traffic Classification Baselines`: We firstly compare against the deep learning baselines, including FS-Net Liu et al. (2019), BiLSTM_Att Yao et al. (2019), Datanet Wang et al. (2018), DeepPacket Lotfollahi et al. (2020), and TSCRNN Lin et al. (2021). Then, we compare against competitive pretraining-based methods, including YaTC Zhao et al. (2023), TrafficLLM Cui et al. (2025), and ET-BERT Lin et al. (2022). (ii) `Traffic Generation Baselines`: For network generation, we compare against the GAN-based Netshare Yin et al. (2022) and the pretraining-based TrafficLLM Cui et al. (2025).

**Evaluation Metrics.** For the network traffic classification tasks, the Accuracy (AC) and Macro F1 Score (F1) are used to assess the performance. For the netowrk traffic generation, the Jensen-Shannon Divergence (JSD) and Total Variation Distance (TVD) are used, following NetGPT Meng et al. (2023)'s experimental setting. The JSD measures how similar two probability distributions are, while the TVD identifies the largest difference in probabilities between two distributions. Lower JSD and TVD indicate better performance.

## 4.1 Network Traffic Classification Performance

We evaluate `Lens`'s traffic classification capabilities against two lines of state-of-the-art approaches: deep learning models and pretraining-based foundation models in Table 2 and 3. All deep learning and pretraining-based models are implemented on open-sourced code and follow the experimental setting as described in their paper. The 12 classification tasks are detailed as below: For ISCX-VPN, we have VPN detection (Task 1), VPN Service detection (Task 2), and VPN application classification (Task 3). For the ISCX-Tor and USTC-TFC-2016, our task involves Tor service detection (Task 4) and USTC-TFC-2016 Application Classification (Task 5). For the Cross Platform (Android), we have an application classification (Task 6) and a country detection task (Task 7). Similarly, for the Cross Platform (IOS), we have application classification (Task 8) and country detection (Task 9). For CIC-DoHBrw-2020(DoH), we include DoH query method classification (Task 10). In the CIC-IoT-2023, we have the IoT attack detection (Task 11) and IoT attack method detection (Task 12). More details about the task and class numbers are detailed in Table 11 in Appendix A.3.

**In both Table 2 and 3, we can see that `Lens` outperforms all baselines with superior accuracy and F1 in 8 out of 12 tasks while performing comparably in the remaining tasks.** The results indicate that pretraining-based models mostly outperform deep learning models, demonstrating the benefits of pretrained network representations. The TrafficLLM Cui et al. (2025) does not perform well, as it suffers from the modality gap between network input and natural language, but also the imbalanced label distribution. This shows the necessity of pre-training network foundation models on network traffic data.

Table 2: Performance on traffic classification from Task 1 to Task 6. `Lens` achieves superior results than baselines, especially on Task 2, 3, and 4. Bold numbers denote the best results, while underlined numbers are the second best.

| Method | Task 1 | | Task 2 | | Task 3 | | Task 4 | | Task 5 | | Task 6 | |
|---|---|---|---|---|---|---|---|---|---|---|---|---|
| | AC | F1 | AC | F1 | AC | F1 | AC | F1 | AC | F1 | AC | F1 |
| FS-Net | 0.9785 | 0.9537 | 0.7360 | 0.6732 | 0.5681 | 0.5910 | 0.9271 | 0.7606 | 0.8074 | 0.8817 | 0.2822 | 0.1219 |
| BiLSTM_Att | 0.9798 | 0.9551 | 0.8009 | 0.7719 | 0.6103 | 0.6635 | 0.9421 | 0.6095 | 0.9463 | 0.9568 | 0.8091 | 0.6023 |
| Datanet | 0.9726 | 0.9405 | 0.7755 | 0.7209 | 0.5762 | 0.5717 | 0.9362 | 0.5483 | 0.9397 | 0.9540 | 0.7081 | 0.4566 |
| DeepPacket | 0.9603 | 0.9178 | 0.7934 | 0.7585 | 0.6137 | 0.6834 | 0.9410 | 0.6405 | 0.9372 | 0.9456 | 0.7993 | 0.5622 |
| TSCRNN | 0.9668 | 0.9233 | 0.7975 | 0.7568 | 0.6028 | 0.6483 | 0.9368 | 0.6049 | 0.9412 | 0.9513 | 0.9072 | 0.8319 |
| YaTC | 0.9834 | 0.9627 | 0.8078 | 0.7805 | 0.6420 | 0.6990 | 0.9362 | 0.6956 | 0.9533 | **0.9707** | 0.9409 | 0.8173 |
| TrafficLLM | 0.9083 | 0.8084 | 0.6757 | 0.4826 | 0.5092 | 0.4650 | 0.9415 | 0.7076 | 0.6598 | 0.6363 | NA | NA |
| ET-Bert | 0.9863 | 0.9692 | 0.8130 | 0.8033 | 0.6484 | 0.6662 | 0.9296 | 0.6336 | 0.9462 | 0.9604 | **0.9800** | **0.8925** |
| Lens (Ours) | **0.9942** | **0.9870** | **0.8979** | **0.8893** | **0.8406** | **0.8137** | **0.9692** | **0.8120** | **0.9538** | 0.9676 | 0.9660 | 0.8847 |

Table 3: Performance on traffic classification from Tasks 7 to 12. `Lens` outperforms baselines significantly on Task 10 and 12, while achieves excellent generalization capabilities. "Avg." is the average accuracy.

| Method | Task 7 | | Task 8 | | Task 9 | | Task 10 | | Task 11 | | Task 12 | | Avg. |
|---|---|---|---|---|---|---|---|---|---|---|---|---|---|
| | AC | F1 | AC | F1 | AC | F1 | AC | F1 | AC | F1 | AC | F1 | AC |
| FS-Net | 0.8552 | 0.5157 | 0.2457 | 0.1052 | 0.3605 | 0.1767 | 0.5275 | 0.1381 | 0.3808 | 0.2758 | 0.9448 | 0.4270 | 0.6345 |
| BiLSTM_Att | 0.9440 | 0.8519 | 0.9080 | 0.8414 | 0.9563 | 0.9567 | 0.9781 | 0.7024 | 0.9809 | 0.9798 | 0.9695 | 0.4283 | 0.9021 |
| Datanet | 0.9141 | 0.7855 | 0.6527 | 0.4508 | 0.9343 | 0.9347 | 0.9742 | 0.6564 | 0.9799 | 0.9788 | 0.9454 | 0.4518 | 0.8591 |
| DeepPacket | 0.9291 | 0.8210 | 0.6330 | 0.3435 | 0.9412 | 0.9414 | 0.9774 | 0.7321 | 0.9814 | 0.9804 | 0.9655 | 0.4714 | 0.8727 |
| TSCRNN | 0.9267 | 0.7953 | 0.9072 | 0.8319 | 0.9510 | 0.9513 | 0.9781 | 0.7024 | 0.9800 | 0.9790 | 0.9698 | 0.4604 | 0.9054 |
| YaTC | 0.9960 | 0.9896 | 0.9552 | 0.9160 | 0.9963 | 0.9962 | 0.9905 | 0.9083 | 0.9864 | 0.9857 | 0.9586 | 0.5885 | 0.9289 |
| TrafficLLM | 0.9473 | 0.8519 | NA | NA | 0.9865 | 0.9864 | 0.4680 | 0.2461 | 0.7927 | 0.7439 | 0.5472 | 0.1844 | 0.7436 |
| ET-Bert | 0.9944 | 0.9855 | **0.9788** | 0.9456 | **0.9988** | **0.9987** | 0.9837 | 0.8151 | 0.9851 | 0.9842 | 0.9455 | 0.4296 | 0.9325 |
| Lens (Ours) | **0.9960** | **0.9898** | 0.9752 | 0.9492 | 0.9951 | 0.9951 | **0.9963** | **0.9610** | **0.9877** | **0.9870** | **0.9878** | **0.6802** | **0.9633** |

On challenging Tasks 2, 3, 4, 10, and 12, `Lens` outperforms all other baselines significantly. In ISCX-VPN datasets (Task 2 and Task3), `Lens` achieves accuracies of 84.06% and 89.79% , representing a +8.49% and +20.64% accuracy improvement over YaTC and ET-BERT separately. The F1 improvement of +11.47% on Task 2 and +8.49% on Task 3 further demonstrate `Lens`'s robust performance on the imbalanced test set. For ISCX-Tor dataset(Task 4), DoH dataset(Task 10), and IoT (Task 12), `Lens` surpasses the second-best baselines significantly by +11.13%, +5.26%, +9.17% on F1, while achieving higher accuracy than baselines on all these tasks. The main reason is that `Lens`'s pretraining on KG-MSP with the context has learned generalizable network representations, better capturing networking semantics. Besides, the context-aware finetuning provides auxiliary task information and label options, finetuning pretrained representations to output labels well.

In Tasks 6 and 8, `Lens` performs slightly better or comparably to baseline methods with both high accuracy and F1, demonstrating excellent generalization capabilities on an unseen dataset, even with 209 classes and 196 classes separately. In contrast, the TrafficLLM deteriorates as finetuning on limited data fails to mitigate the modality gap and learns the large label sets well. Additionally, TrafficLLM can not adapt well to Task 12 due to the skewed test label distribution.

Table 4: `Lens` outperforms in F1 score on head and long-tailed classes in IoT attack method detection.

| Attack Type | Sample Size (n) | ET-BERT | YaTC | Lens (Ours) |
|---|---|---|---|---|
| DDoS | 11055 | 0.973 | 0.979 | **0.995** |
| DoS | 1216 | 0.746 | 0.795 | **0.968** |
| Spoofing | 55 | 0.486 | 0.667 | **0.779** |
| Web-based | 10 | – | – | **0.308** |
| BruteForce | 5 | – | 0.500 | **0.571** |

As shown in Table 4, we provide a detailed case study on Task 12 IoT method detection. `Lens` consistently outperforms the competitive baselines ET-BERT and YaTC across both head and long-tailed classes, with test sample sizes ranging from 5 to 11k. In particular, `Lens` accurately distinguishes DDoS from DoS, and effectively recognizes long-tailed classes such as Web-based and BruteForce, whereas other baselines malfunctioned on these classes.

Table 5: Extensibility performance on Task 6 (209 classes) and Task 8 (196 classes). `Lens` consistently achieves superior extension to unseen classes while maintaining strong accuracy on known classes.

| Scenarios | Task 6 (209 classes) | | | | | | Task 8 (196 classes) | | | | | |
|---|---|---|---|---|---|---|---|---|---|---|---|---|
| | 1 new | | 3 new | | 5 new | | 1 new | | 3 new | | 5 new | |
| | AC | F1 | AC | F1 | AC | F1 | AC | F1 | AC | F1 | AC | F1 |
| ET-BERT | 0.8536 | 0.7832 | 0.6387 | 0.4400 | 0.7137 | 0.5906 | 0.8257 | 0.8026 | 0.8477 | 0.8695 | 0.6079 | 0.5762 |
| ET-BERT-LwF | 0.9378 | **0.8688** | 0.8783 | 0.8625 | 0.8261 | **0.8269** | 0.9211 | **0.8941** | 0.8416 | **0.9174** | 0.7862 | **0.8940** |
| YaTC | 0.7963 | 0.7228 | 0.8066 | 0.7559 | 0.6604 | 0.6039 | 0.7463 | 0.7680 | 0.7598 | 0.7759 | 0.5981 | 0.4361 |
| YaTC-LwF | 0.8459 | 0.7549 | 0.7954 | 0.7510 | 0.6983 | 0.6868 | 0.8583 | 0.8847 | 0.8249 | 0.8897 | 0.8021 | 0.7084 |
| `Lens` (Ours) | **0.9565** | 0.8578 | **0.9518** | **0.8659** | **0.9199** | 0.8264 | **0.9397** | 0.8861 | **0.8962** | 0.8801 | **0.8730** | 0.8407 |

## 4.2 Network Traffic Extensibility Performance

We also evaluate `Lens`'s extensibility on Task 6 with 209 classes and Task 8 with 196 classes. Specifically, we set 3 scenarios where models need to face 1, 3, or 5 novel classes at each time, simulating the real network scenario, where models need to be updated frequently to classify new application types.

To evaluate the extensibility, we first finetune models on old classes until convergence. Then, we finetune only on data samples from new classes to test how well models extend the classification to new classes. Technically, we incorporate new-class options into `Lens` 's finetuning context, whereas ET-BERT and YaTC extends its MLP classifier by adding output dimensions for the new classes. Besides, we further incorporate the learning-without-forgetting (LwF) mechanism Li & Hoiem (2017) into their classifier for a more comprehensive comparison, denoted as ET-BERT-LwF and YaTC-LwF. For both ET-BERT and YaTC, and their LwF variants, the newly added output dimensions of their MLP classifiers are initialized using Kaiming initialization He et al. (2015). All test scenarios include data from both old and new classes to enable fair performance comparison. Detailed experimental settings and results are provided in Appendix C.1.

**As shown in Table 5, we can observe `Lens` extends to new classes well with better accuracy and F1 compared to ET-BERT, YaTC and YaTC-LwF, while always achieves better accuracy than ET-BERT-LwF.** When extending to the challenging 5 new classes, `Lens` outperforms ET-BERT and YaTC significantly with more than +20% improvement of accuracy and F1 score on both tasks. Although both their LwF variants performs much better than the original one, our `Lens` still outperforms with better accuracy. In the 1-new-class scenario, `Lens` performs better than ET-BERT with average +10.85% accuracy and +7.91% F1 gains, and YaTC with average +17.68% accuracy and 12.66% F1 gains. Similarly, on Task 8 with 3 new classes, `Lens` achieves higher accuracy than ET-BERT, YaTC, and their LwF variants and comparable F1 scores compared to ET-BERT, ET-BERT-LwF, and YaTC-LwF.

Nevertheless, on Task 6 with 3 new classes, ET-BERT reallocates logits aggressively toward the new classes, resulting in 31.31% and 42.59% drops in accuracy and F1 compared with `Lens`. ET-BERT-LwF alleviates the drastic shift toward new classes, but it still struggles to balance old and new classes and yields lower overall accuracy. On the other side, YaTC exhibits more stable performance trends regardless of whether with or without the LwF mechanism. The possible reason is that YaTC is built on image-like, fixed-layout traffic representations that induce stronger structural invariance. This contributes to the stable performance, but also reduces the extensibility to new classes. In contrast, our reformulation enables `Lens` to extend to classify new classes more seamlessly via context-aware finetuning, achieving a better balance between old and new classes' performance.

In this paper, we aim to show our model's extensibility to new classes rather than developing new continual learning methods. Although continual learning could further enhance `Lens` 's capability, we leave this for future work. More detailed performance on each class are listed in Table 16, 17, and 18 of Appendix C.1.

Table 6: Performance on traffic generation tasks in terms of JSD and TVD. We can see that `Lens` outperforms baselines consistently on all datasets in generating Destination Port. In source IP generation, `Lens` achieves comparable or better performances than baselines. Bold numbers denote the best results, while underlined numbers are the second best.

| Datasets | Method | JSD ↓ | | | | | TVD ↓ | | | | |
|---|---|---|---|---|---|---|---|---|---|---|---|
| | | Src IP | Dst IP | Src Port | Dst Port | Len | Src IP | Dst IP | Src Port | Dst Port | Len |
| ISCX-VPN | NetShare | 0.3591 | 0.3787 | 0.6539 | 0.5893 | 0.6793 | 0.5802 | 0.5948 | 0.9632 | 0.9137 | 0.9893 |
| | TrafficLLM | **0.0946** | 0.1175 | 0.5742 | 0.0430 | 0.0513 | 0.1849 | 0.1772 | 0.5920 | 0.0600 | 0.0845 |
| | Lens (Ours) | 0.0974 | **0.0905** | **0.5574** | **0.0271** | **0.0338** | **0.1719** | **0.1245** | **0.5789** | **0.0343** | **0.0469** |
| ISCXTor | NetShare | 0.3084 | 0.4160 | 0.5835 | 0.5736 | 0.6531 | 0.4930 | 0.6436 | 0.8813 | 0.8807 | 0.9756 |
| | TrafficLLM | 0.0023 | **0.3629** | **0.5635** | 0.1770 | 0.0359 | 0.0047 | **0.4519** | **0.5838** | 0.2339 | **0.0500** |
| | Lens (Ours) | **0.0022** | 0.4842 | 0.5826 | **0.1337** | 0.0398 | **0.0038** | 0.5620 | 0.6133 | **0.1877** | 0.0560 |
| USTC-TFC | NetShare | 0.4415 | 0.5065 | 0.5731 | 0.5885 | 0.6826 | 0.6876 | 0.7794 | 0.9037 | 0.9114 | 0.9933 |
| | TrafficLLM | **0.3702** | 0.4186 | **0.3496** | 0.2746 | 0.0159 | 0.3915 | **0.4731** | **0.3656** | 0.3261 | 0.0259 |
| | Lens (Ours) | 0.3783 | 0.4361 | 0.3864 | **0.2685** | **0.0143** | **0.3910** | 0.4748 | 0.4076 | **0.2901** | **0.0203** |
| Cross Platform (IOS) | NetShare | 0.3304 | 0.5267 | 0.6056 | 0.6104 | 0.6120 | 0.3389 | 0.5188 | 0.9289 | 0.9318 | 0.9491 |
| | TrafficLLM | **0.0003** | 0.2433 | **0.5784** | 0.0134 | **0.0521** | **0.0006** | 0.3374 | 0.6006 | 0.0265 | **0.0662** |
| | Lens (Ours) | **0.0003** | 0.3241 | 0.6508 | **0.0083** | 0.0608 | **0.0006** | 0.4523 | 0.6746 | **0.0132** | 0.0780 |
| Cross Platform (AN) | NetShare | 0.3459 | 0.3964 | 0.6219 | 0.6269 | 0.6312 | 0.5437 | 0.6205 | 0.9353 | 0.9443 | 0.9598 |
| | TrafficLLM | 0.0016 | **0.1969** | **0.6011** | 0.0082 | **0.0672** | 0.0216 | **0.3101** | 0.6220 | 0.0377 | **0.0835** |
| | Lens (Ours) | **0.0003** | 0.2809 | 0.6531 | **0.0046** | 0.0690 | **0.0041** | 0.4166 | 0.6796 | **0.0130** | 0.0872 |
| CIRA-CIC-DoHBrw | NetShare | 0.3955 | 0.6117 | 0.5446 | 0.4630 | 0.6566 | 0.3886 | 0.6315 | 0.8825 | 0.7754 | 0.9776 |
| | TrafficLLM | 0.0162 | **0.3782** | **0.4858** | **0.0001** | 0.0614 | 0.0838 | **0.4676** | **0.5258** | **0.0003** | 0.1069 |
| | Lens (Ours) | **0.0041** | 0.4105 | 0.6915 | **0.0001** | **0.0481** | **0.0246** | 0.4896 | 0.7065 | **0.0003** | **0.0728** |
| CIC-IoT-2023 | NetShare | 0.0732 | 0.0804 | 0.5939 | 0.1188 | 0.6807 | 0.2155 | 0.2273 | 0.8920 | 0.2936 | 0.9904 |
| | TrafficLLM | 0.0598 | 0.0345 | 0.5544 | 0.0586 | **0.0039** | 0.1523 | 0.1072 | 0.5844 | 0.0833 | 0.0061 |
| | Lens (Ours) | **0.0146** | **0.0098** | **0.0179** | **0.0262** | **0.0039** | **0.0779** | **0.0325** | **0.0217** | **0.0295** | **0.0058** |

## 4.3 Network Traffic Generation Performance

In addition, we evaluate `Lens`'s performance on traffic generation tasks and compare with SOTA baselines, including NetShare Yin et al. (2022), and TrafficLLM Cui et al. (2025). Based on the methodology of NetShare Yin et al. (2022), we generate five vital network header fields: source IP, destination IP, source port, destination port, and packet length at the packet level. The generation of these synthetic header fields can help create usable pcap traffic for network simulation. For each dataset in our experiments, all generation tasks are individually conducted using a supervised finetuning approach. For the metrics JSD and TVD, lower values indicate a smaller distribution difference between the generated and ground-truth data.

From the Table 6, on the generation of destination port, `Lens` outperforms Netshare and TrafficLLM consistently over the 6 datasets. **Lens reduces JSD and TVD up to 55.29% and 64.59%, indicating Lens's significantly better understanding of destination port numbers.** Notably, even `Lens` does not pre-train on the CrossPlatform dataset, it still surpasses the TrafficLLM by +38.06% and +50.19% on JSD and TVD, respectively. This is because our KG-MSP intentionally learns the networking knowledge of port numbers, correlating specific destination port numbers with typical protocol types.

Besides, `Lens` generates more aligned source IP with lower TVD on all datasets, which benefits from the auxiliary context pretraining that correlates dataset sources with network traffic. For source port and destination IP generation, TrafficLLM performs slightly better than `Lens`. This is because the TrafficLLM has more parameters that can memorize randomly assigned source port numbers and varied destination IP addresses better. However, even TrafficLLM has way more parameters, `Lens` excels over TrafficLLM in most packet length generation tasks. Lastly, pretraining-based `Lens` and TrafficLLM perform better than GAN-based Netshare on all tasks. This is because pretraining-based models learn better packet-level network representation, while Netshare sacrifices it for global distribution similarity.

**Network Traffic Generation Fuzzing Tests.** To evaluate the fidelity of the generated network traffic, we conduct network fuzzing tests on the IoT Malicious detection task (Task 11). First, we resume `Lens` from its finetuned checkpoint and generate network header fields using the Task 11 training data. Then, machine learning models such as Decision Tree, SVM and MLP, are trained on the generated network header fields

as a binary classifier to detect malicious network traffic. Lastly, the trained machine learning models are evaluated on the real Task 11 test set for malicious traffic detection, thereby assessing the quality of the synthetic network data.

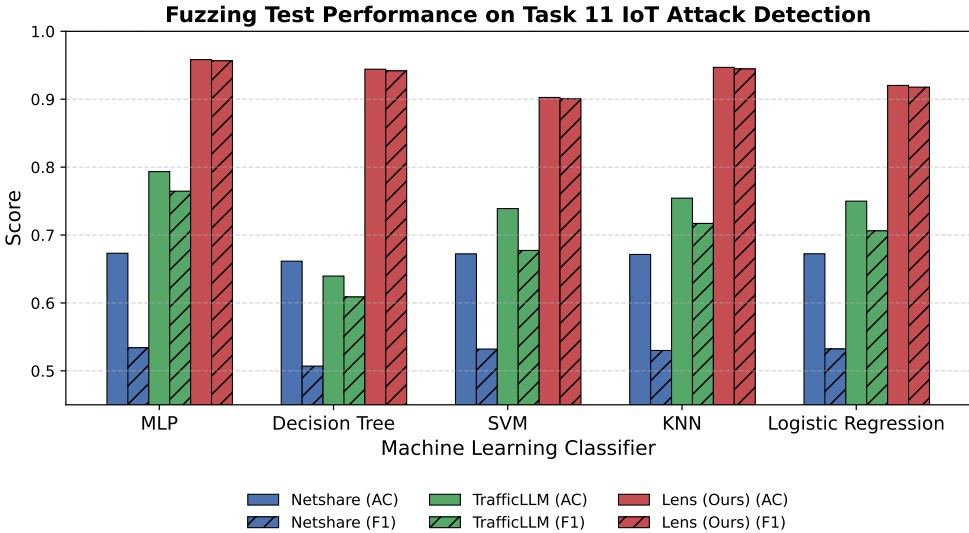

Figure 3: Fuzzing performance on IoT attack detection. Machine-learning models trained on `Lens`-generated traffic achieve consistently higher accuracy and F1 than those trained on baselines' generated traffic.

From Figure 3, we can observe **Lens generates high-fidelity network traffic that aligns better with real scenarios and achieves substantially better performance than baselines on all machine learning-based detectors.** Specifically, `Lens` performs better than the second best TrafficLLM with +30.46% and +33.3% improvement on accuracy and F1. As the payload is usually encrypted, users can either replay saved network payload or generate them randomly with an encrypted algorithm like AES Brown (2023) for a more practical use.

## 4.4 Ablation Studies

**KG-MSP pretraining and context-aware finetuning contribute to classification and generation.**

For network traffic classification, we conduct ablation studies on Tor service detection (Task 4) and VPN application classification (Task 3). In Table 7, both KG-MSP and context-aware finetuning (FT-Context) yield substantial gains, improving F1 by +4.2% and +4.99% on Tor service detection. For VPN application classification, KG-MSP boosts accuracy and F1 by +11.6% and +5.2%, and FT-Context provides an additional +5.25% accuracy improvement. For network traffic generation, we evaluate the VPN destination-port task. As shown in Table 8, FT-Context reduces JSD and TVD by 7.82% and 10.68%, while KG-MSP

Table 7: Ablation studies on Tor service detection and VPN application classification. Both KG-MSP and FT-Context contribute significantly to performance improvement.

| Settings | Tor Service Detection | | VPN App. Classification | |
|---|---|---|---|---|
| | AC | F1 | AC | F1 |
| Lens (Full model) | **0.9692** | **0.8120** | **0.8406** | **0.8137** |
| w/o FT-Context | 0.9577 | 0.7700 | 0.7881 | 0.8111 |
| w/o KG-MSP | 0.9612 | 0.7621 | 0.7246 | 0.7622 |

provides further reductions of 12.8% and 21.5%. Overall, KG-MSP strengthens representation learning by

masking key metadata and payload information, while FT-Context reduces the modality gap during finetuning. These results demonstrate that both KG-MSP and FT-Context are essential for both network traffic classification and generation.

Table 8: Ablation studies on VPN Destination Port generation. Both KG-MSP and FT-Context contribute to better JSD and TVD performance.

| Settings | JSD ↓ | TVD ↓ |
|---|---|---|
| Lens (Full model) | **0.0271** | **0.0343** |
| w/o FT-Context | 0.0294 | 0.0384 |
| w/o KG-MSP | 0.0308 | 0.0437 |

## 5   Discussion and Conclusion

In this paper, we proposed `Lens`, a unified knowledge-guided foundation model for network traffic excelling in both network traffic classification and generation. Through pretraining with Knowledge-guided Mask Span Prediction with textual context, `Lens` effectively learns generalizable network representations from large-scale unlabeled traffic data. To effectively extend classification to new classes, we reframe traffic classification as a closed-ended generation task and handle the distribution shifts of new-class data through context-aware finetuning. Finally, we evaluated the performance of `Lens` on 6 real-world datasets, including 12 traffic classification tasks and 5 network generation tasks. For traffic classification, extensive experimental results demonstrated that `Lens` outperforms the baselines with a large margin in most tasks, showcasing its strong effectiveness and extensibility. For traffic generation, `Lens` generates better high-fidelity network traffic for effective network simulations, outperforming baselines substantially in fuzzing tests. Furthermore, ablation studies validate the effectiveness of KG-MSP and context-aware finetuning. In the future, we plan to implement the proposed method in real-world computer systems to evaluate its performance.

## Acknowledgments

Research reported in this paper was supported in part by NSF CPS-2311086, NSF CIRC-716152, CCI C-3-Q25-WM-02, NAIRR-250288, and the Faculty Research Grant at William & Mary 141446. This work also used the Delta system at the National Center for Supercomputing Applications (NCSA) through ACCESS allocation CIS250781. The ACCESS program is supported by U.S. National Science Foundation grants #2138259, #2138286, #2138307, #2137603, and #2138296.

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

# A More Details on Datasets and Preprocessing

## A.1 Preprocessing Pipeline

Firstly, we segment raw traffic into flows using SplitCap Netresec (2025) based on the standard 5-tuple (source/destination IP, ports, and protocol). Next, we parse each flow with Tshark Wireshark Foundation (2025) to extract the network, transport, and application-layer metadata, and append a 12-byte hexadecimal application payload following NetFound Guthula et al. (2023). Finally, we anonymize all source and destination IP addresses using two special placeholder tokens, consistent with YaTC Zhao et al. (2023), to ensure privacy and prevent identifier leakage.

Table 9: Statistics of packet numbers and tokenized flow lengths of the dataset.

| Dataset | 0 percentile | 25 percentile | 50 percentile | 75 percentile | 100 percentile |
|---|---|---|---|---|---|
| Number of Packets in each flow | 5 | 10 | 21 | **34** | 2,429,639 |
| The Length of tokenized flow | 175 | 567 | 973 | **1,466** | 126,871,264 |

We truncate each flow to its first 34 packets, corresponding to the 75th percentile of the packet-length distribution as shown in Table 9. This choice balances (i) retaining sufficient semantic information contained in early packets (e.g., handshake, protocol negotiation, application identification), and (ii) maintaining a manageable context length for Transformer-based models. Empirically, using 34 packets preserves $>= 75\%$ of classification-relevant signals while keeping the sequence length below 1.5k tokens, which is the maximum context length supported by TurboT5 on our hardware.

## A.2 Dataset Percentage

Table 10: Dataset for Pretraining and Finetuning. The Cross Platform dataset are excluded from pretraining; used for generalization evaluation in finetuning.

| Dataset | Pretraining | Finetuning |
|---|---|---|
| ISCX-VPNDraper-Gil et al. (2016) | 60% | 24% |
| ISCX-TorHabibi Lashkari. et al. (2017) | 60% | 24% |
| USTC-TFC-2016Wang et al. (2017a) | 60% | 5% |
| Cross Platform (Android)Van Ede et al. (2020) | NA | 15% |
| Cross Platform (IOS)Van Ede et al. (2020) | NA | 26% |
| CIC-DoHBrw-2020MontazeriShatoori et al. (2020) | 60% | 6% |
| CIC-IoT-2023Neto et al. (2023) | 60% | 5% |

### A.3 Benchmark Tasks

Table 11: Overview of 12 Classification Tasks Across Datasets

| Task # | Dataset | Task Description | Classes |
|---|---|---|---|
| Task 1 | ISCX-VPN | VPN detection | 2 |
| Task 2 | ISCX-VPN | VPN Service detection | 6 |
| Task 3 | ISCX-VPN | VPN application classification | 16 |
| Task 4 | ISCX-Tor and USTC-TFC-2016 | Tor service detection | 7 |
| Task 5 | USTC-TFC-2016 | Application Classification | 16 |
| Task 6 | Cross Platform (Android) | Application classification | 209 |
| Task 7 | Cross Platform (Android) | Country detection | 3 |
| Task 8 | Cross Platform (iOS) | Application classification | 196 |
| Task 9 | Cross Platform (iOS) | Country detection | 3 |
| Task 10 | CIC-DoHBrw-2020 (DoH) | DoH query method classification | 5 |
| Task 11 | CIC-IoT-2023 | IoT attack detection | 2 |
| Task 12 | CIC-IoT-2023 | IoT attack method detection | 7 |

## B  Ablation and Analysis of Pretraining Objective

### B.1  Comparison with Random Masking Strategies

**KG-MSP consistently outperforms random masking during pretraining.** As shown in Table 12, `Lens` achieves +1.91% / +8.6% higher accuracy and +3.26% / +4.05% higher F1 on Task 2 and Task 3, respectively. This improvement stems from masking protocol-critical metadata and payload-related information rather than arbitrary spans. Since random masking here corresponds to the MASS-style/T5-style span masking objective Raffel et al. (2020), these results highlight the benefit of our knowledge-guided design.

Table 12: Ablation study of pretraining strategy with KG-MSP or Random Masking. Models pretrained with KG-MSP perform better than random masking.

| Pretraining | Task 2 | | Task 3 | |
|---|---|---|---|---|
| | AC | F1 | AC | F1 |
| Random Masking | 0.8788 | 0.8567 | 0.7546 | 0.7732 |
| KG-MSP(Ours) | **0.8979** | **0.8893** | **0.8406** | **0.8137** |

### B.2  Ablation of isolated model components

**KG-MSP contributes more than Contextual Input among pretraining component designs.** We further conduct ablation studies to isolate the contribution of each component. (i) Knowledge-guided masked span pretraining (KG-MSP), (ii) Contextual Input design, and (iii) classification-as-generation (CLS-AS-GEN). We want to clarify that the row "w/o Contextual Input and KG-MSP" in Table 13 and Table 14 corresponds to direct fine-tuning without any pretraining, same as the row "w/o KG-MSP" in Table 7.

As shown in Table 13 and Table 14, on Task 4, KG-MSP adds **+3.23%** F1 versus **+1.76%** F1 from Contextual Input; On Task 3, KG-MSP improves **+6.18%** AC / **+3.48%** F1 versus **+5.42%** AC / **+1.67%** F1 from Contextual Input. Across the four results, KG-MSP yields larger gains than Contextual Input, confirming that the knowledge-guided pretraining objective is the primary factor.

**CLS-AS-GEN is introduced mainly for extensibility rather than accuracy.** It unifies classification and generation within a single encoder-decoder structure and **allows Lens to extend to new classes while preserving knowledge of existing ones**. We include its ablation here for completeness. To

Table 13: Ablation studies on Task 3 (VPN application classification).

| Settings | AC | F1 |
|---|---|---|
| **Lens (Full model)** | **0.8406** | **0.8137** |
| w/o CLS-AS-GEN | 0.7662 | 0.7808 |
| w/o Contextual Input | 0.7864 | 0.7970 |
| w/o Contextual Input and KG-MSP | 0.7246 | 0.7622 |

Table 14: Ablation studies on Task 4 (Tor service detection).

| Settings | AC | F1 |
|---|---|---|
| **Lens (Full model)** | **0.9692** | **0.8120** |
| w/o CLS-AS-GEN | 0.9689 | 0.8076 |
| w/o Contextual Input | 0.9670 | 0.7944 |
| w/o Contextual Input and KG-MSP | 0.9612 | 0.7621 |

ablate it, we use mean pooling over encoder outputs followed by a linear head, following SentenceT5 Ni et al. (2022). CLS-AS-GEN improves **+0.44%** F1 on Task 4, and **+7.44%** AC/**+3.29%** F1 on Task 3. The larger improvement on Task 3 is due to two factors. First, the VPN Application classification task (Task 3) involves finer-grained categories (e.g., distinguishing between Email and Gmail classes) and noisier network traffic, where Lens' decoder's cross-attention needs to selectively attend to informative parts of the input. Second, Lens is pretrained with a coupled **encoder-decoder objective** that requires the decoder to reconstruct masked network-specific spans. Classifying with the encoder alone discards the decoder pathway and underuses the pretrained signal, which is more visible on harder tasks.

### B.3  Sensitivity Analysis of Masking Configuration

Table 15a shows that `Lens` achieves its best validation performance when the masking ratio $\theta$ is 60%, and overall remains stable across different ratios. Similarly, Table 15b indicates that masking Seq/Ack/Length with ratio $k$ also leads to minimal performance variation. This insensitivity arises because KG-MSP defaults to random masking when fewer than 15% of tokens are masked. For generality, we therefore adopt a 50% masking ratio.

Table 15: Sensitivity Analysis on Masking Vital Network Metadata (left) and Seq/Ack/Length (right).

(a) Analysis of $\theta$ for Masking Vital Metadata

| Mask Prob. | 30% | 60% | 90% |
|---|---|---|---|
| | AC | AC | AC |
| `Lens` (Ours) | 0.33 | **0.34** | 0.32 |

(b) Analysis of $k$ for Masking Seq/Ack/Length

| Mask Prob. | 25% | 50% | 75% |
|---|---|---|---|
| | AC | AC | AC |
| `Lens` (Ours) | **0.35** | **0.35** | 0.34 |

## C   More detail on Experimental Results

### C.1   Performance of per-class in the extensibility of classification

We summarize the setup of the extensibility experiments. For Task 6, the new classes are selected based on their higher sample counts: com.ifeng.news2 (226) for the 1-class setting; additionally, sohu.sohuvideo (200) and xunlei.downloadprovider (190) for the 3-class setting; and further qiyi.video (150) and youku.phone (150) for the 5-class setting. For Task 8, we follow the same criterion: pocket-pool (416) for the 1-class setting; plus aiqiyi (238) and color-ballz (224) for the 3-class setting; and further yy (206) and youku (182) for the 5-class setting. These labels provide clearer performance differences due to their larger test sets.

Across Table 16, 17, and 18, `Lens` consistently outperforms all baselines on both tasks. ET-BERT fine-tunes new classes with a learning rate of 2e-5, whereas `Lens` uses 5e-6 due to architectural differences, illustrating `Lens`'s stronger extensibility and reduced need for task-specific adjustment.

Table 16: Performance of 1 new class in Task 6 and Task 8.

(a) Task 6 — com.ifeng.news2

| Methods | P | R | F1 |
|---|---|---|---|
| ET-BERT | 0.0474 | **1.0000** | 0.0900 |
| **Lens** | **0.9636** | 0.9408 | **0.9521** |

(b) Task 8 — pocket-pool

| Methods | P | R | F1 |
|---|---|---|---|
| ET-BERT | 0.0643 | **1.0000** | 0.1208 |
| **Lens** | **0.6753** | 1.0000 | **0.8062** |

Table 17: Performance of 3 new classes in Task 6 and Task 8.

Task 6

| Methods | com.ifeng.news2 | | | sohu.sohuvideo | | | xunlei.downloadprovider | | |
|---|---|---|---|---|---|---|---|---|---|
| | P | R | F1 | P | R | F1 | P | R | F1 |
| ET-BERT | 0.0484 | **0.9941** | 0.0924 | 0.2213 | **0.9858** | 0.3615 | 0.5930 | 0.9440 | 0.7284 |
| **Lens** | **0.8195** | **0.9941** | **0.8984** | **0.9577** | 0.9645 | **0.9611** | **0.9191** | **1.0000** | **0.9579** |

Task 8

| Methods | pocket-pool | | | aiqiyi | | | color-ballz | | |
|---|---|---|---|---|---|---|---|---|---|
| | P | R | F1 | P | R | F1 | P | R | F1 |
| ET-BERT | 0.1098 | **1.0000** | 0.1978 | 0.0995 | 0.9888 | 0.1809 | 0.5685 | 0.9881 | 0.7217 |
| **Lens** | **0.8041** | **1.0000** | **0.8914** | **0.4120** | **1.0000** | **0.5836** | **0.7000** | **1.0000** | **0.8235** |

Table 18: Performance of 5 new classes in Task 6 and Task 8.

Task 6

| Methods | com.ifeng.news2 | | | sohu.sohuvideo | | | xunlei.downloadprovider | | | qiyi.video | | | youku.phone | | |
|---|---|---|---|---|---|---|---|---|---|---|---|---|---|---|---|
| | P | R | F1 | P | R | F1 | P | R | F1 | P | R | F1 | P | R | F1 |
| ET-BERT | 0.0572 | **1.0000** | 0.1083 | 0.2945 | **0.9858** | 0.4535 | 0.4237 | **1.0000** | 0.5952 | 0.3839 | **1.0000** | 0.5548 | 0.5170 | 0.9681 | 0.6741 |
| **Lens** | **0.7249** | 0.9822 | **0.8342** | **0.8528** | 0.9858 | **0.9145** | **0.8389** | 1.0000 | **0.9124** | **0.7532** | 1.0000 | **0.8592** | **0.6462** | 0.8936 | **0.7500** |

Task 8

| Methods | pocket-pool | | | aiqiyi | | | color-ballz | | | yy | | | youku | | |
|---|---|---|---|---|---|---|---|---|---|---|---|---|---|---|---|
| | P | R | F1 | P | R | F1 | P | R | F1 | P | R | F1 | P | R | F1 |
| ET-BERT | 0.2188 | **1.0000** | 0.3590 | 0.2607 | 0.9551 | 0.4096 | **0.6484** | 0.9881 | **0.7830** | 0.4444 | 0.9870 | 0.6129 | 0.0000 | 0.0000 | 0.0000 |
| **Lens** | **0.8254** | **1.0000** | **0.9043** | **0.5733** | **0.9663** | **0.7197** | 0.6336 | 0.9881 | 0.7721 | **0.5385** | **1.0000** | **0.7000** | **0.6476** | **1.0000** | **0.7861** |

**C.2 Performance of few-shot extensibility in low-resource scenarios**

To evaluate `Lens`'s extensibility to unseen classes in low-resource scenarios, we extend `Lens` to 5 unseen classes with only 5 and 10 labeled examples per class on two application classification tasks (Task 8: CrossPlatform-IOS; Task 6: CrossPlatform-Android). We compare against four representative baselines: ET-BERT and YaTC, together with their Learning-without-Forgetting (LwF) variants that mitigate catastrophic forgetting via KL-divergence regularization.

Table 19: Few-shot performance of 5 unseen classes on Task 8 (CrossPlatform-IOS-APP Classification)

| Models | 5-shot ACC | 5-shot F1 | 10-shot ACC | 10-shot F1 |
|---|---|---|---|---|
| ET-BERT | 0.7129 | 0.6595 | 0.7374 | 0.6759 |
| ET-BERT-LwF | 0.7691 | 0.7058 | 0.7345 | 0.6558 |
| YaTC | 0.8066 | 0.7321 | 0.8131 | 0.7016 |
| YaTC-LwF | 0.8607 | 0.8443 | 0.8962 | 0.8248 |
| **Lens (Ours)** | **0.9548** | **0.9243** | **0.9296** | **0.8914** |

Table 20: Few-shot performance of 5 unseen classes on Task 6 (CrossPlatform-Android-APP Classification)

| Models | 5-shot ACC | 5-shot F1 | 10-shot ACC | 10-shot F1 |
|---|---|---|---|---|
| ET-BERT | 0.8070 | 0.6978 | 0.7204 | 0.5633 |
| ET-BERT-LwF | 0.8682 | 0.6647 | 0.835 | 0.7346 |
| YaTC | 0.6934 | 0.5753 | 0.6883 | 0.5333 |
| YaTC-LwF | 0.8042 | 0.7301 | 0.7398 | 0.6108 |
| **Lens (Ours)** | **0.9267** | **0.8259** | **0.9181** | **0.8234** |

In both Table 19 and 20, `Lens` consistently and substantially outperforms all baselines across both datasets and both shot settings. On Task 8, `Lens` improves over the second-best baseline by **+9.41%** AC / **+8.00%** F1 (5-shot) and **+3.34%** AC / **+6.66%** F1 (10-shot). On Task 6, the gains are **+5.85%** AC / **+9.58%** F1 (5-shot) and **+8.31%** AC / **+8.88%** F1 (10-shot).

Firstly, both ET-BERT and YaTC without LwF fail to balance old and new classes well. They achieve acceptable performance on new classes at the cost of degraded accuracy on old ones. Furthermore, their LwF variants partly address this via KL-divergence regularization and extend to new classes gradually, but still perform worse than our `Lens` due to the distribution shifts in the MLP-based classification heads as new classes are introduced. Secondly, `Lens` reframes the classification as a **closed-ended generation task**, learning label distributions at the decoder side instead of a fixed MLP head. This aligns directly with the extensibility motivation to mitigate the learning-forgetting conflicts in MLP heads.

**C.3 Analysis of `Lens`'s trade-off between learning and forgetting**

We conducted ablation experiments to show (i) `Lens`'s trade-off between extensibility to new classes and maintaining old classes' performance in 3 and 5 unseen class scenarios, and (ii) `Lens`'s forgetting of old classes when extended with 3 versus 5 new classes.

To better quantify `Lens`' trade-off on both new and old classes, we use the harmonic mean (HM) of old class and new class performance, defined as $HM = 2 * (Old * New)/(Old + New)$. HM penalizes methods that sacrifice one side for the other.

**Trade-off between old and new classes.** We evaluate `Lens` under both 3 and 5 unseen classes scenarios on Task 8 (CrossPlatform-IOS Application Classification) and Task 6 (CrossPlatform-Android Application Classification) against two LwF baselines.

Table 21: The performance of the 3-unseen classes on Task 8 (CrossPlatform-IOS-APP Classification)

| Method | Old ACC | Old F1 | New ACC | New F1 | HM-ACC | HM-F1 |
|---|---|---|---|---|---|---|
| ET-BERT-LwF | **0.9600** | **0.9254** | 0.6201 | 0.3720 | 0.7539 | 0.5308 |
| YaTC-LwF | 0.9168 | 0.8612 | 0.8116 | 0.2840 | 0.8610 | 0.4274 |
| **Lens (Ours)** | 0.8801 | 0.8819 | **1.0000** | **0.7662** | **0.9362** | **0.8200** |

Table 22: The performance of the 5-unseen class on Task 8 (CrossPlatform-IOS-APP Classification)

| Method | Old ACC | Old F1 | New ACC | New F1 | HM-ACC | HM-F1 |
|---|---|---|---|---|---|---|
| ET-BERT-LwF | **0.9258** | **0.8662** | 0.6730 | 0.2685 | 0.7793 | 0.4099 |
| YaTC-LwF | 0.8966 | 0.8360 | 0.9409 | 0.5889 | **0.9182** | 0.6911 |
| **Lens (Ours)** | 0.8446 | 0.8424 | **0.9916** | **0.7764** | 0.9122 | **0.8080** |

Table 23: The performance of the 3-unseen class on Task 6 (CrossPlatform-Android-APP Classification)

| Method | Old ACC | Old F1 | New ACC | New F1 | HM-ACC | HM-F1 |
|---|---|---|---|---|---|---|
| ET-BERT-LwF | 0.8889 | 0.7106 | 0.5954 | 0.2812 | 0.7131 | 0.4028 |
| YaTC-LwF | 0.8674 | 0.6954 | 0.7264 | 0.1394 | 0.7907 | 0.2323 |
| **Lens (Ours)** | **0.9479** | **0.8648** | **0.9862** | **0.9391** | **0.9667** | **0.9004** |

Table 24: The performance of the 5-unseen class on Task 6 (CrossPlatform-Android-APP Classification)

| Method | Old ACC | Old F1 | New ACC | New F1 | HM-ACC | HM-F1 |
|---|---|---|---|---|---|---|
| ET-BERT-LwF | **0.9172** | 0.7705 | 0.6065 | 0.3880 | 0.7302 | 0.5161 |
| YaTC-LwF | 0.8054 | 0.6105 | 0.6389 | 0.2785 | 0.7126 | 0.3824 |
| **Lens (Ours)** | 0.9098 | **0.8257** | **0.9768** | **0.8541** | **0.9421** | **0.8397** |

Across all four settings in Table 21, 22, 23, and 24, `Lens` achieves the best HM-F1 (ranging from **80.80%** to **90.04%**) and comparable best HM-ACC (ranging from **91.22%** to **96.67%**). The LwF baselines preserve old-class performance at the cost of poor new class adaptation. For instance, ET-BERT-LwF's new-class F1 falls in the range of 0.2685 to 0.3880, and YaTC-LwF's new-class F1 reaches only 0.1394 on Task 6 (3-Unseen). However, `Lens` extends well to new classes, achieving new-class F1 up to 0.9391 on Task 6. At the same time, `Lens` retains competitive old-class accuracy and F1, yielding the most balanced performance between old and new classes across both tasks and scenarios. This is because we reframe the classification as a closed-ended generation task to mitigate the distribution shift from new classes.

**Lens's forgetting of old classes when extended with 3 versus 5 new classes.** To directly show `Lens`'s forgetting, we compare `Lens`'s old-class performance when the number of newly added classes increases from 3 to 5 on the same task. As shown in Table 25, `Lens`'s old-class performance drops by only 3–4% on both ACC and F1, demonstrating that `Lens` is stable on old classes when extended with new classes, and does not exhibit catastrophic forgetting. We attribute this stability to our reframing of classification as a generation task at the decoder, which accommodates new classes while preserving learned representations of old ones.

Table 25: `Lens`'s old-class performance when extended with 3 vs. 5 new classes.

| Task | Old ACC (3→5) | Δ ACC | Old F1 (3→5) | Δ F1 |
|---|---|---|---|---|
| Task 8 | 0.8801 → 0.8446 | −3.55% | 0.8819 → 0.8424 | −3.95% |
| Task 6 | 0.9479 → 0.9098 | −3.81% | 0.8648 → 0.8257 | −3.91% |

## D   Examples of Model Input

---

**Traffic Classification Data Example**

**Input:** <SIP> → <DIP> TCP 60 34665 → 443 [SYN] Seq=0 Win=29200 Len=0 MSS=1460 SACK_PERM TSval=4177987 TSecr=0 WS=128
<DIP> → <SIP> TCP 60 443 → 34665 [SYN, ACK] Seq=0 Ack=1 Win=42540 Len=0 MSS=1350 SACK_PERM TSval=1083957698 TSecr=4177987 WS=128
<SIP> → <DIP> TCP 52 34665 → 443 [ACK] Seq=1 Ack=1 Win=29312 Len=0 TSval=4177994 TSecr=1083957698
<SIP> → <DIP> TLSv1 569 Client Hello 1603 0102 0001 0001 fc03 03a5
<DIP> → <SIP> TCP 52 443 → 34665 [ACK] Seq=1 Ack=518 Win=43648 Len=0 TSval=1083957724 TSecr=4177994
<DIP> → <SIP> TLSv1.2 201 Server Hello, Change Cipher Spec, Encrypted Handshake Message 1603 0300 5d02 0000 5903 0355
<SIP> → <DIP> TCP 52 34665 → 443 [ACK] Seq=518 Ack=150 Win=30336 Len=0 TSval=4178001 TSecr=1083957725
<SIP> → <DIP> TLSv1.2 103 Change Cipher Spec, Encrypted Handshake Message 1403 0300 0101 1603 0300 2800
<SIP> → <DIP> TLSv1.2 139 Application Data 1703 0300 5200 0000 0000 0000
<SIP> → <DIP> TCP 1390 [TCP segment of a reassembled PDU] 1703 0306 a000 0000 0000 0000
<SIP> → <DIP> TLSv1.2 415 Application Data ce79 99f6 3ab9 6c90 3ddd 5620
<SIP> → <DIP> TLSv1.2 1103 Application Data 1703 0304 1600 0000 0000 0000
<DIP> → <SIP> TLSv1.2 108 Application Data 1703 0300 3300 0000 0000 0000
<DIP> → <SIP> TLSv1.2 94 Application Data 1703 0300 2500 0000 0000 0000 ......
<sep> In aim email facebook ftps gmail hangout icq netflix scp sftp skype spotify torrent vimeo voipbuster youtube, the VPN network traffic belong to:
**Labels:** Hangout

---

**Traffic Generation Data Example**

**Input:** <SIP> → 93.189.89.83 TCP 322 POST /iWEab%20&%26%3DC/kXi_6r+j/Tb HTTP/1.1 [TCP segment of a reassembled PDU] 504f 5354 202f 6957 4561 6225 <sep> Generate source ip for the USTC-TFC2016 network traffic:
**Labels:** <SIP>10.0.2.108

---

Figure 4: The example input of classification and generation. For classification, the input includes parsed network traffic and a task context listing label options. For generation, the input contains a masked packet and a task description.

## E   Hyperparameter Setting

Table 26: Hyperparameters for `Lens`.

|              | Hyperparameter                    | Value            |
|--------------|-----------------------------------|------------------|
|              | Transformer Encoder Layer number  | 12               |
|              | Transformer Decoder Layer number  | 12               |
|              | Attention heads number            | 12               |
| Architecture | Attention heads dimension         | 64               |
|              | Hidden dimension                  | 768              |
|              | MLP hidden                        | 2048             |
|              | MLP activation                    | Gated-GeLU       |
|              | Total gradient steps              | 780k             |
|              | Batch size                        | 48               |
|              | Learning rate                     | $1 \times 5^{-4}$ |
| Pre-training | Dropout rate                      | 0.1              |
|              | Optimizer                         | AdamW            |
|              | Scheduler                         | Warmup cosine    |
|              | Grad clip norm                    | 1.0              |
|              | Scheduler warmup steps            | 13k              |
|              | Total Max epochs                  | 40               |
|              | Batch size                        | 32               |
|              | Learning rate                     | $1 \times 5^{-5}$ |
| Fine-tuning  | Dropout rate                      | 0.1              |
|              | Optimizer                         | AdamW            |
|              | Grad clip norm                    | 1.0              |
|              | Scheduler                         | Warmup linear    |

# F    Ablation of Model Architecture

As encoder-only models like ET-BERT Lin et al. (2022) and YaTC Zhao et al. (2023) only excel in network traffic classification, and decoder-only models pretrained with next token prediction struggle with the network generation due to the auto-regressive nature. Thus, we chose the encoder-decoder architecture because it better captures the global information of the input, which aligns well with our network generation tasks.

Specifically, the network traffic generation requires the model to generate header fields conditioned on the encrypted payload. Our encoder-decoder architecture handles this naturally: the encoder learns global context from the full input (Masked header + payload), and the decoder generates the target header fields, aligning with the Knowledge-guided masked span prediction objective used in pretraining. In contrast, the decoder-only model would need to reverse input order, processing the payload first before predicting header fields, which contradicts the left-to-right auto-regressive pretraining and disrupts learned token dependencies.

We have conducted ablation studies on the model architecture. We pretrain a decoder-only baseline with the same decoder configuration as `Lens`'s decoder, using the next-token prediction task on the same pretraining dataset and format as `Lens`. Then, we finetune the decoder-only model on the network traffic generation tasks with the same settings as `Lens`. From the Table 27, we can tell `Lens` outperforms the decoder-only model across all network traffic generation tasks on the VPN and Tor datasets significantly. This is because `Lens`'s encoder-decoder architecture aligns pretraining and fine-tuning very well, and the encoder learns a better representation of the input network for the decoder. By contrast, the decoder-only model suffers from order mismatch between pretraining and fine-tuning.

Table 27: Ablation study of model architecture on traffic generation tasks in terms of JSD and TVD. `Lens` outperforms decoder-only model on VPN and Tor datasets across 5 generation tasks significantly.

| Datasets | Method | JSD ↓ | | | | | TVD ↓ | | | | |
|---|---|---|---|---|---|---|---|---|---|---|---|
| | | Src IP | Dst IP | Src Port | Dst Port | Len | Src IP | Dst IP | Src Port | Dst Port | Len |
| ISCX-VPN | Decoder-only | 0.6873 | 0.6724 | 0.6201 | 0.0799 | 0.2370 | 0.7055 | 0.6920 | 0.6389 | 0.1012 | 0.3270 |
| | Lens (Ours) | **0.0974** | **0.0905** | **0.5574** | **0.0271** | **0.0338** | **0.1719** | **0.1245** | **0.5789** | **0.0343** | **0.0469** |
| ISCXTor | Decoder-only | 0.1605 | 0.6896 | 0.6921 | 0.3661 | 0.1258 | 0.2757 | 0.7058 | 0.7065 | 0.3991 | 0.1719 |
| | Lens (Ours) | **0.0022** | **0.4842** | **0.5826** | **0.1337** | **0.0398** | **0.0038** | **0.5620** | **0.6133** | **0.1877** | **0.0560** |

# G   Zero-shot Cloze Test on Network Semantics

To test whether knowledge-guided pretraining captures better network-specific semantics, we conducted the zero-shot cloze tests to probe `Lens`'s knowledge about port-protocol associations, protocol flags, and packet length, compared with a baseline pretrained with random masking pretraining (RandMSP).

**Port-protocol associations.** We masked the source and destination ports in 2k validation network packets for zero-shot prediction. For example, we mask "56574 → 443" of network traffic with "<extra_id_0>" to predict the destination port 443, as the destination port 443 is associated with the HTTPS protocol. We use beam search with a width of 5 and report models' top-1 and top-5 soft match accuracy (i.e., top-5 is counted as correct if the ground-truth port appears anywhere in the top-5 beam outputs).

Table 28: The accuracy of the zero-shot cloze test on destination port number

| Method | top-1 | top-5 |
|---|---|---|
| RandMSP | 0.1940 | 0.2510 |
| **Lens (Ours)** | **0.8470** | **0.8800** |

In Table 28, we observe `Lens` outperforms RandMSP significantly in understanding the associations between destination port numbers and network protocols because of the knowledge-guided pretraining mechanism. Specifically, for common protocols like HTTPS(443), HTTP(80), and HTTP-alt(8080), `Lens` achieved top-1 accuracy of 98.4%, 93.4%, and 93.7% on each protocol, respectively, compared with 36.9%, 27.1%, and 45% for RandMSP. Besides, for less common protocols like BGP(179), IRC(6667), `Lens` still gained accuracy of 71.4% and 93.9%, while RandMSP failed to produce any correct prediction.

**TCP protocol flags.** Following the same procedure, we evaluate zero-shot recovery of TCP flags. In

Table 29: The accuracy of the zero-shot cloze test on TCP protocol flag

| Method | top-1 | top-5 |
|---|---|---|
| RandMSP | 0.2490 | 0.2790 |
| **Lens (Ours)** | **0.3600** | **0.4220** |

Table 29, the results indicate `Lens` achieved **+11.1%** and **+14.3%** accuracy gain on top-1 and top-5 predictions, respectively. This demonstrates `Lens`'s better understanding of the correlation between TCP flags and connection-level intentions, including synchronization, acknowledgment, and connection termination. Specifically, `Lens` predicts "SYN" with 88.2% top-1 accuracy versus 79.2% for RandMSP. Furthermore, `Lens` excels at "FIN, RST" (91.9%) and "PSH, ACK" (41.2%), while RandMSP failed almost entirely on these flags due to its bias towards the dominant "ACK" flag.

**Packet length.** We further examine packet length, which correlates with network application types. For example, DNS queries are typically 70 to 90B, TLS Client Hellos are around 200 to 600B. As predicting exact packet length is inherently noisy, we divide packet length into different buckets as S(<80B), M(80-250B), L(250-800B), and XL(800+B). As shown in Table 30, `Lens` outperforms RandMSP by **+32.5%** and **+38.5%**

Table 30: The accuracy of the zero-shot cloze test on packet length (bucket match)

| Method | Bucket Match Top-1 | Bucket Match Top-5 |
|---|---|---|
| RandMSP | 9.3% | 16.4% |
| **Lens (Ours)** | **41.8%** | **54.9%** |

on top-1 and top-5 bucket match, respectively. In detail, for protocols like FTP, DNS, and STUN, `Lens` performs with an accuracy of 50%, 59.1%, and 92.6%, compared with 2%, 3.9%, and 5.6% for RandMSP.

