# OpenReview forum: "Lens: A Knowledge-Guided Foundation Model for Network Traffic"
_TMLR — Accepted by TMLR_

### Review · Reviewer_6oJi · 2026-02-02

**Summary Of Contributions:**

The paper proposes Lens, a foundation model for network traffic prediction. It is based on an encoder-decoder architecture, namely T5 combined with knowledge guided most span prediction in pretraining. In contrast to previous works, classification tasks are considered as a special case of generation problems during training, in order to provide better results at adapting to the distribution shift occured. This architecture is then evaluated on generation and classification tasks with significantly improved results compared to previous encoder or decoder models.

**Additional Comments:**

Questions:

1. Why was T5 chosen as the encoder-decoder architecture instead of other architectures? Especially since T5 is a rather old architecture to begin with?

2. Furthermore, what drives the choice of an encoder-decoder model opposed to previous approaches

3. On page 10 "...This is because the TrafficLLM
has more parameters that can memorize randomly assigned source port numbers and varied destination IP
addresses better...". Why was the parameter count of LENS not increased to alleviate this advantage of TrafficLLM? And why was the parameter count set to 0.25 billion?

4. How does LENS compare in runtime and memory requirements compared to other models?

5. What is the impact of the knowledge guided training? What happens if it is removed from training?

**Audience:**

Yes

**Audience Explanation:**

Although, network traffic prediction is a rather niche topic, the improved results over previous methods should garner some interest from TMLR audience. Furthermore, foundation model claims are supported by good results across datasets, allowing for a sophisticated model across tasks. In addition a well-known approach is used in a new setting, allowing for readers from more LLM aligned domains to potentially show interest.

**Broader Impact Concerns:**

There are no broader impact concerns applicable here.

**Claims And Evidence:**

Yes

**Claims Explanation:**

The underlying architecture is explained well and with T5 a well-known base architecture. Furthermore, claims about empirical results are validated in the experimental section of the paper. Since the paper mostly proposes a new architecture with empirical results there are no theoretical results to evaluate. However, all parts of the architecture are provided through related works and seen in other models or applications.
Since hyperparameters, pretraining datasets and model specifications are given it is possible to replicate results. It would be great to additionally provide the code used for the experiments.

**Requested Changes:**

Major: There are in my opinion no major issues with the current paper.


Minor:
1. The paper would benefit from a more thorough discussion of the selected architecture and how it differs from related works. The current section showcases the architecture already but does not highlight the differences.

2. In terms of writing it would be great to have the knowledge based masking highlighted more.

3. There is a typo in the section title "experiments".

4. For generalization tasks it would be beneficial to have additional baselines to compare LENS with. In contrast classification results have substantially more baselines as comparison.

5. The paper would further benefit from ablation studies on how specific parts of the proposed architecture improve performance.

---

> ### Author Response · Authors · 2026-03-11
>
> Dear Reviewer 6oJi,
>
> We sincerely thank you for taking the time to review our paper and for providing insightful and constructive feedback. We have carefully addressed all the points and suggestions you raised. Please feel free to let us know if you have any additional questions or concerns. We greatly appreciate the opportunity to further refine our work. Thank you again for your thoughtful review and valuable input.
>
> **Q1&2: Why was T5 chosen as the encoder-decoder architecture instead of other architectures? Furthermore, what drives the choice of an encoder-decoder model opposed to previous approaches?**
>
> **A1&2:** The reason why we choose the encoder-decoder-based architecture is that **it better captures the global information of the input data, which aligns well with our network traffic generation tasks.** For example, the network traffic generation requires the model to generate header fields conditioned on the encrypted payload. Our encoder-decoder architecture handles this naturally: the encoder learns global context from the full input (Masked header + payload), and the decoder generates the target header fields, aligning with the Knowledge-guided masked span prediction objective used in pretraining. In contrast, the decoder-only model would need to reverse input order, processing the payload first before predicting header fields, which contradicts the left-to-right auto-regressive pretraining and disrupts learned token dependencies.
>
> Besides, we conducted an architecture ablation study updated in Appendix F to compare with the decoder-only baseline. In detail, we pretrain a decoder-only baseline with the same decoder configuration as Lens’s decoder (without the cross-attention module). For generality, we pretrain the decoder-only baselines via the next token prediction task on the same pretraining dataset and format as Lens. Then, we also finetune the decoder-only model on the network traffic generation tasks with the same settings as Lens.
>
> From the results in **Table 1 to 4**, Lens outperforms the decoder-only baseline significantly on 5 network traffic generation tasks on both VPN and Tor datasets. **This validates the effectiveness of our encoder-decoder architecture choice, which better aligns with our pretraining and downstream tasks, and the encoder learns a better representation of the input network for the decoder.** Contrastingly, the decoder-only model pretrained with the next token prediction suffers from the order mismatch of pretraining and fine-tuning.
>
> **Table 1: The architecture impact on JSD performance of VPN Generation tasks**
> | Baseline |Src IP (JSD &darr;)|Dst IP (JSD &darr;)|Src Port (JSD &darr;)|Dst Port (JSD &darr;)|Packet Length  (JSD &darr;)|
> |---------------------------------------|----------------------------------|--------|----------------------------------------|--------|------------------------------|
> |Decoder-only  |0.6873|	0.6724|	0.6201|	0.0799|	0.237|
> |LENS|**0.0974**|**0.0905**|**0.5574**|**0.0271**|**0.0338**|
>
> **Table 2: The architecture impact on TVD performance of VPN Generation tasks**
> | Baseline |Src IP (TVD &darr;)|Dst IP (TVD &darr;)|Src Port (TVD &darr;)|Dst Port (TVD &darr;)|Packet Length  (TVD &darr;)|
> |---------------------------------------|----------------------------------|--------|----------------------------------------|--------|------------------------------|
> |Decoder-only|0.7055|0.692|0.6389|0.1012|0.327|
> |LENS|**0.1719**|**0.1245**|**0.5789**|**0.0343**|**0.0469**|
>
> **Table 3: The architecture impact on JSD performance of Tor Generation tasks**
> | Baseline |Src IP (JSD &darr;)|Dst IP (JSD &darr;)|Src Port (JSD &darr;)|Dst Port (JSD &darr;)|Packet Length  (JSD &darr;)|
> |---------------------------------------|----------------------------------|--------|----------------------------------------|--------|------------------------------|
> |Decoder-only  |0.1605| 0.6896|	0.5826 |0.3661 |	0.1258|
> |LENS|**0.0022**| **0.4842**| **0.5826**|**0.1337**|**0.0398**|
>
> **Table 4: The architecture impact on TVD performance of Tor Generation tasks**
> | Baseline |Src IP (TVD &darr;)|Dst IP (TVD &darr;)|Src Port (TVD &darr;)|Dst Port (TVD &darr;)|Packet Length  (TVD &darr;)|
> |---------------------------------------|----------------------------------|--------|----------------------------------------|--------|------------------------------|
> |Decoder-only|0.2757|0.7058|0.6133 |0.3991|0.1719|
> |LENS|**0.0038**|**0.562**|**0.6133**|**0.1877**|**0.056**|
>
> **Q3: Why was the parameter count of LENS not increased to alleviate this advantage of TrafficLLM? And why was the parameter count set to 0.25 billion?**
>
> **A3:** The model size of our Lens is selected based on the size of our pretraining datasets and the tradeoff of performance and efficiency. **With 0.25B model parameters, our model has already performed well enough on downstream tasks compared to larger models.**

---

> > ### Author Response · Authors · 2026-03-11
> >
> > **Q4: How does LENS compare in runtime and memory requirements compared to other models?**
> >
> > **A4:** The deployment of LENS needs 4.57GB of memory, and the inference achieves 27.33 samples/second without optimization on an A6000 GPU. Besides, TrafficLLM based on GLM-6B (6B, \~12GB)  has an inference time of 4.33 samples/second on an A6000 GPU, which has way lower efficiency than Lens. While BERT (\~110M parameters, \~2GB) and YaTC(\~1.86M parameters, \~112MB) offer higher throughput, they generalize poorly to unseen classes and can not conduct network traffic generation tasks. Similarly, traditional deep learning baselines require fewer resources but demonstrate inferior performance.
> >
> > **Q5: What is the impact of the knowledge-guided training? What happens if it is removed from training?**
> >
> > **A5:** As we have demonstrated the effectiveness of the knowledge-guided pretraining in Table 8 of our ablation study. **Compared with the vanilla randon masking, Lens achieved **+11.6%** accuracy gain and **+5.2%** F1 gain on the VPN Application Classification, and **+4.99%** F1 increase on the Tor Service Detection with knowledge-guided pretraining.**
> >
> > **Q6: The paper would benefit from a more thorough discussion of the selected architecture and how it differs from related works. The current section showcases the architecture already but does not highlight the differences.**
> >
> > **A6:** We have updated a thorough discussion and conducted further ablation study of model architecture in Appendix F. **Lens outperforms the decoder-only model on all network traffic generation tasks significantly across VPN and Tor datasets.**
> >
> > **Q7: In terms of writing it would be great to have the knowledge based masking highlighted more.**
> >
> > **A7:** We have highlighted the knowledge-guided MSP in the introduction of our revised manuscript as following, "we propose the Knowledge-Guided Mask Span Prediction (KG-MSP) to intentionally mask network metadata and payload-related information as a whole based on their importance in networking. Specifically, learning representations of port numbers and protocols as indivisible semantic units is critical for application classification, while understanding protocol flags and key fields as complete entities is essential for interpreting packet intentions and protocol behaviors."
> >
> > **Q8: There is a typo in the section title "experiments".**
> >
> > **A8:** We have fixed it in our revised manuscript.

---

> > > ### Author Response · Authors · 2026-03-11
> > >
> > > **Q9: For generalization tasks it would be beneficial to have additional baselines to compare LENS with other baselines.**
> > >
> > > **A9:** **We have added another baseline YaTC's performance compared to our models.** Here, we didn't add the TrafficLLM because it fails on large-scale label classification tasks (196 and 209 classes) due to the limited training data for each class. Besides, other classical deep learning based models are not added, as they are feature-based, while our Lens is pre-training based.
> > >
> > > **From Table 5 to 6, LENS still achieves the best overall performance among all baselines in terms of the generalization capabilities.** YaTC exhibits relatively more stable performance trends than ET-BERT with or without LwF mechanism, especially on CrossPlatform Android Application Classifications. A possible explanation is that YaTC’s image-like, fixed-layout traffic representation induces stronger structural invariance in the learned features. While this contributes to stable performance, it may also hurt to adapt to the distribution shifts introduced by novel classes, thereby limiting the additional benefits of LwF. We have already updated the tables in our updated manuscript.
> > >
> > > **Table 5: Extensibility performance on CrossPlatform-Android-APP-CLS**
> > > | Scenarios | 1 novel class (AC) | 1 novel class (F1) | 3 novel classes (AC) | 3 novel classes (F1) | 5 novel classes (AC) | 5 novel classes (F1) |
> > > |:---|:---|:---|:---|:---|:---|:---|
> > > | ET-BERT | 0.8536 | 0.7832 | 0.6387 | 0.44 | 0.7137 | 0.5906 |
> > > | ET-BERT-LwF | 0.9378 | **0.8688** | 0.8783 | 0.8625 | 0.8261 | **0.8269** |
> > > |YaTC|0.7963|0.7228|0.8066|0.7559|0.6604|0.6039|
> > > |YaTC-LwF|0.8459|0.7549|0.7954|0.751|0.6983|0.6868|
> > > | Lens(ours) | **0.9565** | 0.8578 | **0.9518** | **0.8659** | **0.9199** | 0.8264 |
> > >
> > > **Table 6: Extensibility performance on CrossPlatform-IOS-APP-CLS**
> > > | Scenarios | 1 novel class (AC) | 1 novel class (F1) | 3 novel classes (AC) | 3 novel classes (F1) | 5 novel classes (AC) | 5 novel classes (F1) |
> > > |:---|:---|:---|:---|:---|:---|:---|
> > > | ET-BERT | 0.8257 | 0.8026 | 0.8477 | 0.8695 | 0.6079 | 0.5762 |
> > > | ET-BERT-LwF | 0.9211 | **0.8941** | 0.8416 | **0.9174** | 0.7862 | **0.894** |
> > > |YaTC|0.7463|0.7680|0.7598|0.7759|0.5981|0.4361|
> > > |YaTC-LwF|0.8583|0.8847|0.8249|0.8897|0.8021|0.7084
> > > | Lens(ours) | **0.9397** | 0.8861 | **0.8962** | 0.8801 | **0.8730** | 0.8407 |
> > >
> > > **Q10: The paper would further benefit from ablation studies on how specific parts of the proposed architecture improve performance.**
> > >
> > > **A10:** We have conducted the architecture ablation study compared to the decoder-only model, and updated in our Appendix F. Lens's encoder-decoder design outperforms the decoder-only model greatly, which demonstrates the importance of the encoder in our pretraining and finetuning tasks. Furthermore, in Table 7 and 8 of our paper, we demonstrate the contribution of finetuning context and Knowledge-guided MSP on both classification and generation tasks. The importance of knowledge-guided masking is further demonstrated by comparing with random masking in Table 12 (Appendix B), and the robustness of different masking ratios is ablated in Table 13 (Appendix B).

---

### Review · Reviewer_qkkY · 2026-02-15

**Summary Of Contributions:**

The paper proposes Lens, a T5-style encoder–decoder foundation model for both network traffic classification and traffic header generation. The main contributions are:
1.  Instead of random token masking, Lens introduces knowledge-guided masking that masks field-level functional units, aiming to better preserve the semantics of protocol fields during pretraining.
2.  The paper reframes traffic classification as a closed-ended generation task by placing label options in the prompt. This design avoids relying on a fixed MLP head and enables incremental extension to new classes via prompt updates and lightweight finetuning on new-class data.
3.  Lens is evaluated on 12 classification tasks and 5 generation tasks. It outperforms baselines on the majority of benchmarks, and the generation quality is further validated via fuzzing tests.

**Audience:**

Yes

**Audience Explanation:**

1. It provides a concrete example of injecting protocol structure into pretraining via field-level, knowledge-guided masking.
2.  The closed-ended generation formulatio is a practical alternative to fixed-head classifiers when the label space expands incrementally.
3.  The traffic header generation component could be useful for security workflows such as data augmentation and fuzzing-style evaluation.

**Broader Impact Concerns:**

There is no ethical concern in this paper.

**Claims And Evidence:**

Yes

**Claims Explanation:**

1. Lens reports an average accuracy of ~96.3% across 12 benchmark tasks, and achieves the best results on 8 out of 12 tasks among the compared baselines.
2. In the incremental new-class setting on Task 6 and Task 8, Lens achieves substantially better Accuracy/F1 than ET-BERT, and is consistently no worse in Accuracy than ET-BERT-LwF.
3. For generation, the paper uses JSD/TVD to quantify distributional similarity between generated and real traffic, and further validates usefulness through fuzzing tests: classifiers trained on Lens-generated traffic generalize better to real test data than those trained on baseline-generated traffic.
4. The ablations suggest that KG-MSPprovides measurable gains over random masking, and that context-aware finetuning is important for both classification and generation performance.

**Requested Changes:**

1. The paper introduces masking hyperparameters (e.g., $\theta$ and $k$) for specific metadata. While a sensitivity study appears in the appendix, I suggest adding one or two sentences in the main text explaining the rationale behind the chosen values and how sensitive the method is to different masking ratios.
2. Since context-aware finetuning requires listing label options in the prompt, please discuss scalability when the label set grows to hundreds/thousands: (i) input-length limits, and (ii) potential interference when many labels share prefixes or are semantically similar (which could bias decoding).
3. Lens processes only the first 34 packets of a flow (capped at 1.5k tokens). Please clarify whether this truncation harms tasks where discriminative patterns appear later in long-lived flows, and whether performance degrades as flow length increases.
4. Please elaborate on the role of the appended natural-language descriptions: do you observe consistent gains over a prompt that contains only parsed/hex fields and label options? An ablation comparing with vs. without the natural-language descriptions would clarify whether explicit text–field alignment is necessary.

---

> ### Author Response · Authors · 2026-03-11
>
> Dear Reviewer qkkY,
>
> We sincerely thank you for taking the time to review our paper and for providing insightful and constructive feedback. We have carefully addressed all the points and suggestions you raised. Please feel free to let us know if you have any additional questions or concerns. We greatly appreciate the opportunity to further refine our work. Thank you again for your thoughtful review and valuable input.
>
> **Q1: The paper introduces masking hyperparameters (e.g., $\theta$ and $k$) for specific metadata. While a sensitivity study appears in the appendix, I suggest adding one or two sentences in the main text explaining the rationale behind the chosen values and how sensitive the method is to different masking ratios.**
>
> **A1:** We added 2 sentences to explain the insights and robustness of hyperparameter selection in the updated manuscript. **(i).** "To prioritize learning these important fields’ representations, Lens assigns higher masking probabilities with θ = 60%. The sensitivity analysis (Table 13a, Appendix B.2) also confirms the robustness to different θ values." **(ii).** We conducted a sensitivity analysis on k (Table 13b, Appendix B.2), which shows robustness to different ratios. We adopt k = 50% to pretrain more on these significant fields, enhancing Lens’s fundamental understanding of protocol behavior.
>
> **Q2: Since context-aware finetuning requires listing label options in the prompt, please discuss scalability when the label set grows to hundreds/thousands: (i) input-length limits, and (ii) potential interference when many labels share prefixes or are semantically similar (which could bias decoding).**
>
> **A2:** To solve the scalability issue, such as in Task 6 (209 classes) and Task 8 (196 classes), **we mapped textual labels into numeric indices, such as denoting "qiyi.video" as 0, "color-ballz" as 1, to keep the context compact and avoid semantic interference between similar labels in practice.** This strategy effectively resolves both concerns: **(i)** it significantly reduces input length as label sets grow, and **(ii)** numeric indices eliminate prefix overlap and semantic similarity, ensuring unambiguous decoding. Combined with our prefix trie[1] constrained decoding, this approach scales to large label sets reliably. We also added additional details in the updated manuscript.
>
> [1] Nicola et al. "Autoregressive entity retrieval". ICLR 2021
>
> **Q3: Lens processes only the first 34 packets of a flow (capped at 1.5k tokens). Please clarify whether this truncation harms tasks where discriminative patterns appear later in long-lived flows, and whether performance degrades as flow length increases.**
>
> **A3:** **Extending beyond 34 packets may yield diminishing returns for the current benchmarks.** For encrypted traffic classification (e.g., VPN application identification), prior work has shown that discriminative features, such as packet size distributions and TLS negotiation fingerprints, are concentrated in early packets[1]. Similarly, for attack detection like IoT attack detection, attack signatures such as scanning probes and brute-force attempts manifest in the initial phase of the attack flow[2].
>
> Besides, the 34-packet truncation is based on dataset statistics (Table 9, Appendix A.1), where 75% of flows contain 34 or fewer packets, meaning the majority are captured without information loss. Among the remaining 25% that exceed 34 packets, the dominant categories are bulk-transfer protocols (e.g., SMB, FTP), whose application-layer identity is established during the initial connection handshake, well within the first 34 packets.
>
> Moreover, tasks involving inherently long-lived flows, such as VPN service detection (Task 2) covering streaming, file transfer, and P2P traffic, still achieve strong performance (89.79% accuracy, 88.93% F1), demonstrating that **the first 34 packets capture sufficient discriminative information even for long-duration flows**."
>
> We acknowledge that certain network traffic with delayed behavioral signatures may benefit from a longer context window. **Our truncation is a practical design choice rather than an architectural constraint.** Our backbone supports longer sequences, making it straightforward to extend the context window for such scenarios in future work.
>
> [1] Husák M et al. HTTPS traffic analysis and client identification using passive SSL/TLS fingerprinting[J]. EURASIP Journal on Information Security, 2016.
>
> [2] Korba et al. "Ai-driven fast and early detection of iot botnet threats: A comprehensive network traffic analysis approach." IWCMC 2024.

---

> > ### Author Response · Authors · 2026-03-11
> >
> > **Q4: Please elaborate on the role of the appended natural-language descriptions: do you observe consistent gains over a prompt that contains only parsed/hex fields and label options? An ablation comparing with vs. without the natural-language descriptions would clarify whether explicit text–field alignment is necessary.**
> >
> > **A4:** **We ablated LENS with and without natural-language descriptions on network classification tasks, and observed consistent gains of text-field alignment.** We detail the necessity of text-field alignment in two folds as follows.
> >
> > **(i) For downstream network classification**, the results in the following Tables 1-3 show that the natural-language descriptions improve the performance. More importantly, our complete text-field in our classification tasks includes both the **natural-language description** and **label options** for the overall significant performance gains. According to Table 8 in our submission, removing the whole text-field leads to substantial degradation on Tasks 3 with 5.17% F1 loss and 4 with 6.25% accuracy degradation, which further confirms the importance of text-field alignment.
> >
> > Furthermore, **(ii) for downstream network generation**, the natural language description plays a more critical role. The generation prompt only contains the natural language description, such as "Generate Source IP for the VPN network traffic" without label options. As shown in Table 4 (Table 7 of our manuscript), **removing the natural language description decreased the performance by 8.49% and 11.95% on JSD and TVD separately**, demonstrating the necessity for guiding the generation process.
> >
> >
> > Table 1: The performance of T2 VPN Service Classification
> > |                       | ACC    | Ma-F1  |
> > |-----------------------|--------|--------|
> > | Lens(Full Model)      | **0.8979** | **0.8893**|
> > | W/o NLP Prompt        | 0.8892 |  0.8751  |
> >
> > Table 2: The performance of T3 VPN Application Classification
> > |                       | ACC    | Ma-F1  |
> > |-----------------------|--------|--------|
> > | Lens(Full Model)      | **0.8406** | 0.8137|
> > | W/o NLP Prompt        | 0.8308 | **0.8161**|
> >
> > Table 3: The performance of T4 Tor Service Detection
> > |                       | ACC    | Ma-F1  |
> > |-----------------------|--------|--------|
> > | Lens(Full Model)      | **0.9692**|**0.8120**|
> > | W/o NLP Prompt        | 0.9649 |0.8069|
> >
> > Table 4: The performance of VPN Destination Port Generation (Table 7 in the manuscript)
> > |                       | JSD &darr;|TVD &darr; |
> > |-----------------------|--------|--------|
> > | Lens(Full Model)      | **0.0271**|**0.0343**|
> > | W/o FT-Context        | 0.0294 |0.0384|
> > | W/o KG-MSP            | 0.0308 |0.0437|

---

### Review · Reviewer_XG1G · 2026-04-17

**Summary Of Contributions:**

This paper proposes Lens, a unified knowledge-guided foundation model for network traffic for both traffic classification and traffic generation. The model is built on an encoder-decoder T5 architecture and combines Knowledge-Guided Mask Span Prediction (KG-MSP) with context-aware finetuning. It evaluates Lens on 12 classification tasks and 5 generation tasks across 6 public datasets, and reports strong empirical results on both performance and extensibility. The main strengths are the unified formulation, the network-aware pretraining design, and the broad experimental study. A main limitation is that the proposed knowledge-guided component is mainly instantiated through masking design and textual context, so the paper focuses more on effective network-specific modeling and empirical gains than on deeper forms of explicit knowledge integration.

**Audience:**

Yes

**Audience Explanation:**

Some ML readers, especially those interested in domain-specific foundation models or applied sequence modeling, may find it interesting.

**Broader Impact Concerns:**

One potential ethical concern is that the paper supports network traffic generation in addition to classification, which means the method could in principle be used to generate more realistic synthetic traffic for certain purposes, not only for simulation or testing.

**Claims And Evidence:**

Yes

**Claims Explanation:**

Partially. The empirical results are fairly broad and show the method works well in practice, but some of the stronger claims, especially around the knowledge-guided aspect and new-class extensibility, are not fully discussed.

**Requested Changes:**

1. The paper should better justify the current knowledge-guided framing and align this positioning more closely with the actual method. At present, the method mainly relies on manually designed masking rules for selected network fields together with short textual context, which appears closer to network-aware pretraining than to a stronger form of explicit knowledge integration.
2. The paper should provide more direct evidence that the proposed design captures network-specific semantics. For example, it would be useful to test whether the model better understands port–protocol associations, protocol flags, or other field-level regularities.
3. The author should consider stronger baselines based on general seq2seq or LLM-style models. Since the method relies heavily on textual context and generation-style prediction, the current experiments do not fully establish why a dedicated network-specific foundation model is necessary, rather than simply helpful.
4. The paper should better isolate the source of the reported gains. Currently, network-aware masking, contextual input design, and classification-as-generation are introduced together, so it is difficult to tell which component brings the improvement and to what extent.
5. The experiments should include few-shot or other low-resource new-class settings. Emerging applications or attack types often come with very limited labeled data.
6. The paper should analyze the trade-off between adapting to new classes and retaining performance on old classes. Although the paper compares against LwF-style baselines, it does not provide enough analysis of whether Lens itself suffers from forgetting or instability during incremental updates.

---

> ### Author Response · Authors · 2026-04-23
>
> Dear Reviewer XG1G,
>
> We sincerely thank you for taking the time to review our paper and for providing insightful and constructive feedback. We have carefully addressed all the points and suggestions you raised. Please feel free to let us know if you have any additional questions or concerns. We greatly appreciate the opportunity to further refine our work. Thank you again for your thoughtful review and valuable input.
>
> **Q1: The paper should better justify the current knowledge-guided framing and align this positioning more closely with the actual method. At present, the method mainly relies on manually designed masking rules for selected network fields together with short textual context, which appears closer to network-aware pretraining than to a stronger form of explicit knowledge integration.**
>
> **A1:** We have reframed our knowledge-guided framing in our introduction. For clarification, knowledge-guided pretraining in our manuscript means integrating domain knowledge into the model pretraining process, rather than relying on generic self-supervised signals.
>
> Concretely, we draw insights from how human experts analyze network traffic, such as identifying applications through port numbers and protocols, understanding packet intentions via flags, and analyzing protocol behaviors through sequence numbers. Then, we design the network knowledge-aware masking mechanism to learn better representations of critical network terms and protocols during pretraining.
>
> The effectiveness of our pretraining method is supported empirically in Table 1 (Table 12 of the manuscript). Our knowledge-guided masking outperforms random masking on downstream classification tasks, with an average of **+5.25%** accuracy gain and **+3.65%** F1 gain, indicating the necessity of the knowledge-guided design.
>
> **Table 1: The ablation study of pretraining strategy with KG-MSP or Random Masking.**
> | Pretraining | Task 2 AC | Task 2 F1 | Task 3 AC | Task 3 F1 |
> |---|---|---|---|---|
> | Random Masking | 0.8788 | 0.8567 | 0.7546 | 0.7732 |
> | **KG-MSP (Ours)** | **0.8979** | **0.8893** | **0.8406** | **0.8137** |
>
> More broadly, we view our proposed methodology as a general recipe for designing domain-specific foundation pretraining methods, where structured expert knowledge is available.

---

> > ### Author Response · Authors · 2026-04-23
> >
> > **Q2: The paper should provide more direct evidence that the proposed design captures network-specific semantics. For example, it would be useful to test whether the model better understands port–protocol associations, protocol flags, or other field-level regularities.**
> >
> > **A2:** To provide direct evidence, we conducted zero-shot cloze tests to probe whether Lens captures network-specific semantics more effectively than a baseline pretrained with random masking pretraining (RandMSP).
> >
> > **1）Port-protocol associations.** We masked the source and destination ports in 2k validation network packets for zero-shot prediction. For example, we mask "56574 → 443" of network traffic with "<extra\_id\_0>" to predict the destination port 443, as the destination port 443 is associated with the HTTPS protocol. We use beam search with a width of 5 and report their top-1 and top-5 soft match accuracy (i.e., top-5 is counted as correct if the ground-truth port appears anywhere in the top-5 beam outputs)
> >
> > **Table 2: The accuracy of the zero-shot cloze test on destination port number**
> > | Method | top-1 | top-5 |
> > | :--- | :---: | :---: |
> > | RandMSP | 0.1940 | 0.2510 |
> > | **Lens(Ours)** | **0.8470** | **0.8800** |
> >
> > In Table 2, we observe Lens outperforms RandMSP significantly in understanding the associations between destination port numbers and network protocols because of the knowledge-guided pretraining mechanism. Specifically, for common protocols like HTTPS(443), HTTP(80), and HTTP-alt(8080), Lens achieved top-1 accuracy of 98.4%, 93.4%, and 93.7% on each protocol, respectively, compared with 36.9%, 27.1%, and 45% for RandMSP. Besides, for less common protocols like BGP(179), IRC(6667), Lens still gained accuracy of 71.4% and 93.9%, while RandMSP failed to produce any correct prediction.
> >
> > **2）TCP protocol flags.** Following the same procedure, we evaluate zero-shot recovery of TCP flags.
> >
> > **Table 3: The accuracy of the zero-shot cloze test on TCP protocol flag**
> > | Method | top-1 | top-5 |
> > | :--- | :---: | :---: |
> > | RandMSP | 0.2490 | 0.2790 |
> > | **Lens(Ours)** | **0.3600** | **0.4220** |
> >
> > In Table 3, the results indicate Lens achieved **+11.1%** and **+14.3%** accuracy gain on top-1 and top-5 predictions, respectively. This demonstrates Lens's better understanding of the correlation between TCP flags and connection-level intentions, including synchronization, acknowledgment, and connection termination. Specifically, Lens predicts "SYN" with 88.2% top-1 accuracy versus 79.2% for RandMSP. Furthermore, Lens excels at "FIN,RST"(91.9%) and "PSH,ACK"(41.2%), while RandMSP failed almost entirely on these flags due to its bias towards the dominant "ACK" flag.
> >
> > **3）Packet length.** We further examine packet length, which correlates with network application types. For example, DNS queries are typically 70 to 90B, TLS Client Hellos are around 200 to 600B. As predicting exact packet length is inherently noisy, we divide packet length into different buckets as S(<80B), M(80-250B), L(250-800B) , and XL(800+B).
> >
> > **Table 4: Zero-shot cloze test on packet length (bucket match)**
> > | Method | Bucket Match Top-1 | Bucket Match Top-5 |
> > | :--- | :---: | :---: |
> > | RandMSP | 9.3% | 16.4% |
> > | **Lens (Ours)** | **41.8%** | **54.9%** |
> >
> > As shown in Table 4, Lens outperforms RandMSP by **+32.5%** and **+38.5%** on top-1 and top-5 bucket match, respectively. In detail, for protocols like FTP, DNS, and STUN, Lens performs with an accuracy of 50%, 59.1%, and 92.6%, compared with 2%, 3.9%, and 5.6% for RandMSP.
> >
> > **Summary.** Based on the three zero-shot cloze tests on destination port number, TCP protocol flags, and Packet length, we can tell Lens captures the network-specific semantics substantially better than RandMSP because of our specific knowledge-guided pretraining mechanism. We have included these results in the appendix of the revised manuscript.

---

> > > ### Author Response · Authors · 2026-04-23
> > >
> > > **Q3: The author should consider stronger baselines based on general seq2seq or LLM-style models. Since the method relies heavily on textual context and generation-style prediction, the current experiments do not fully establish why a dedicated network-specific foundation model is necessary, rather than simply helpful.**
> > >
> > > **A3:** We have compared with the SOTA baseline TrafficLLM, a strong LLM-style baseline built on ChatGLM2-6B. In Tables 2, 3, and 6 of our manuscript, Lens outperforms TrafficLLM on both network traffic classification and generation tasks. We list the performance comparison here.
> > >
> > > **Classification (Tables 5–6).** Across 12 downstream classification tasks, Lens achieves an average accuracy of **96.33%**, compared with **74.36%** for TrafficLLM. We attribute TrafficLLM's degradation primarily to (i) the modality gap between the network traffic input and natural language pretraining, and (ii) sensitivity to imbalanced label distributions inherent to traffic classification. However, our Lens is pretrained on network traffic from scratch with a knowledge-guided objective and aligns both the textual context and the network traffic input well.
> > >
> > > **Table 5: The performance on traffic classification from Task 1 to Task 6.**
> > >
> > > | Method       | Task 1 AC   | Task 1 F1   | Task 2 AC   | Task 2 F1   | Task 3 AC   | Task 3 F1   | Task 4 AC   | Task 4 F1   | Task 5 AC   | Task 5 F1     | Task 6 AC     | Task 6 F1     |
> > > | ------------ | ----------- | ----------- | ----------- | ----------- | ----------- | ----------- | ----------- | ----------- | ----------- | ------------- | ------------- | ------------- |
> > > | TrafficLLM   | 0.9083      | 0.8084      | 0.6757      | 0.4826      | 0.5092      | 0.4650      | 0.9415      | 0.7076      | 0.6598      | 0.6363        | NA            | NA            |
> > > | Lens (Ours)  | **0.9942**  | **0.9870**  | **0.8979**  | **0.8893**  | **0.8406**  | **0.8137**  | **0.9692**  | **0.8120**  | **0.9538**  | **0.9676** | **0.9660** | **0.8847** |
> > >
> > > **Table 6: The performance on traffic classification from Tasks 7 to 12.**
> > >
> > > | Method       | Task 7 AC  | Task 7 F1  | Task 8 AC     | Task 8 F1  | Task 9 AC  | Task 9 F1  | Task 10 AC | Task 10 F1 | Task 11 AC | Task 11 F1 | Task 12 AC | Task 12 F1 | Avg. AC    |
> > > | ------------ | ---------- | ---------- | ------------- | ---------- | ---------- | ---------- | ---------- | ---------- | ---------- | ---------- | ---------- | ---------- | ---------- |
> > > | TrafficLLM   | 0.9473     | 0.8519     | NA            | NA         | 0.9865     | 0.9864     | 0.4680     | 0.2461     | 0.7927     | 0.7439     | 0.5472     | 0.1844     | 0.7436     |
> > > | Lens (Ours)  | **0.9960** | **0.9898** | **0.9752** | **0.9492** | **0.9951**    | **0.9951**     | **0.9963** | **0.9610** | **0.9877** | **0.9870** | **0.9878** | **0.6802** | **0.9633** |
> > >
> > >
> > > **Network Generation (Table 7).** We summarize Lens vs. TrafficLLM 's generation performance comparison (Table 6 in our manuscript) on 5 tasks across 7 datasets based on JSD (↓) and TVD(↓) in the following Table 7.
> > >
> > > Lens outperforms TrafficLLM on **destination port**, **packet length**, and **source IP**, which are most related to protocol semantics, application types, and dataset-level context. On destination IP and source port, TrafficLLM is often better, which we attribute to the nature of these fields: **client-side source ports are ephemerally assigned, and destination IPs are highly diverse across deployments.** As these fields have little generalizable structure, TrafficLLM with 6B parameters tends to memorize the empirical distribution. This pattern directly supports that **our knowledge-guided pretraining learns better network protocol-level regularities rather than raw field distributions**, demonstrating the necessity of pretraining a network-specific foundation model.
> > >
> > > **Table 7: The performance comparison of Lens and TrafficLLM on 5 generation tasks across 7 datasets via JSD and TVD metrics.**
> > > | Field | Semantic Structure | Lens wins | TrafficLLM wins | Ties |
> > > |---|:---:|:---:|:---:|:---:|
> > > | **Destination Port** | Strong (protocol-related) | **12 / 14** | 0 / 14 | 2 / 14 |
> > > | **Packet Length** | Strong (app-type-related) | **9 / 14** | 3 / 14 | 2 / 14 |
> > > | **Source IP** | Moderate (dataset-related) | **9 / 14** | 5 / 14 | 0 / 14 |
> > > | Destination IP | Weak (highly diverse) | 3 / 14 | 11 / 14 | 0 / 14 |
> > > | Source Port | Weak (ephemeral) | 4 / 14 | 9 / 14 | 1 / 14 |

---

> > > > ### Author Response · Authors · 2026-04-23
> > > >
> > > > **Q4: The paper should better isolate the source of the reported gains. Currently, network-aware masking, contextual input design, and classification-as-generation are introduced together, so it is difficult to tell which component brings the improvement and to what extent.**
> > > >
> > > > **A4:** We have conducted further ablation studies to further isolate the contribution of each component. (i) Knowledge-guided masked span pretraining (KG-MSP), (ii) contextual input design, and (iii) classification-as-generation (CLS-AS-GEN).
> > > >
> > > > We want to clarify that the isolated effect of KG-MSP versus random masking is compared in Table 8 of the manuscript (Table 1 in Q1). The row "w/o Contextual Input and KG-MSP" corresponds to direct fine-tuning without any pretraining. As shown in Table 8 and Table 9, among the pretraining component designs, **KG-MSP contributes more than contextual input**. On Task 4, KG-MSP adds **+3.23%** F1 versus **+1.76%** F1 from Contextual Input; On Task 3, KG-MSP improves **+6.18% AC** / **+3.48% F1** versus **+5.42% AC** / **+1.67%** F1 from Contextual Input. Across the four results, KG-MSP provides larger gains than Contextual Input, confirming knowledge-guided pretraining objective is the primary factor.
> > > >
> > > > **Table 8: Ablation studies on Task 3 (VPN application classification).**
> > > >
> > > > | Settings | AC | F1 |
> > > > | :--- | :---: | :---: |
> > > > | **Lens (Full model)** | **0.8406** | **0.8137** |
> > > > | w/o CLS-AS-GEN | 0.7662 | 0.7808 |
> > > > | w/o Contextual Input | 0.7864 | 0.7970 |
> > > > | w/o Contextual Input and KG-MSP | 0.7246 | 0.7622 |
> > > >
> > > > **Table 9: Ablation studies on Task 4 (Tor service detection).**
> > > >
> > > > | Settings | AC | F1 |
> > > > | :--- | :---: | :---: |
> > > > | **Lens (Full model)** | **0.9692** | **0.8120** |
> > > > | w/o CLS-AS-GEN | 0.9689 | 0.8076 |
> > > > | w/o Contextual Input | 0.9670 | 0.7944 |
> > > > | w/o Contextual Input and KG-MSP | 0.9612 | 0.7621 |
> > > >
> > > > **CLS-AS-GEN is introduced mainly for extensibility rather than accuracy**: it unifies classification and generation within a single encoder-decoder structure and **allows Lens to extend to new classes while preserving knowledge of existing ones.** We include its ablation here for completeness. To ablate it, we use mean pooling over encoder outputs followed by a linear head, following SentenceT5 [1]. CLS-AS-GEN improves **+0.44%** F1 on Task 4, and **+7.44%** AC/**+3.29%** F1 on Task 3. The larger improvement on Task 3 is because of two factors. First, the VPN Application classification task (Task 3) involves finer-grained categories (e.g., distinguishing between Email and Gmail classes), and noisier network traffic, where Lens' decoder's cross-attention needs to selectively attend to informative parts of the input. Second, Lens is pretrained with a coupled **encoder-decoder objective** that requires the decoder to reconstruct masked network-specific spans. Classifying with the encoder alone discards the decoder pathway and underuses the pretrained signal, which is more visible on harder tasks.
> > > >
> > > > [1]Ni, Jianmo, et al. "Sentence-t5: Scalable sentence encoders from pre-trained text-to-text models." ACL 2022.

---

> > > > > ### Author Response · Authors · 2026-04-23
> > > > >
> > > > > **Q5: The experiments should include few-shot or other low-resource new-class settings. Emerging applications or attack types often come with very limited labeled data.**
> > > > >
> > > > > **A5:** We evaluated Lens in the few-shot setting which extends to 5 low-resource unseen classes with only **5 and 10 labeled examples per class** across two application tasks(Task 8: CrossPlatform-iOS; Task 6: CrossPlatform-Android). We compare against four representative baselines: ET-BERT and YaTC, together with their Learning-without-Forgetting (LwF) variants that mitigate catastrophic forgetting via KL-divergence regularization.
> > > > >
> > > > >
> > > > > **Table 10: Few-shot performance of 5 unseen classes on Task 8 (CrossPlatform-IOS-APP Classification)**
> > > > >
> > > > > | Models    | 5-shot ACC | 5-shot F1 | 10-shot ACC | 10-shot F1 |
> > > > > | ------------ | ------------- | ------------ | -------------- | ------------- |
> > > > > | ET-BERT      | 0.7129        | 0.6595       | 0.7374         | 0.6759        |
> > > > > | ET-BERT-LwF  | 0.7691        | 0.7058       | 0.7345         | 0.6558        |
> > > > > | YaTC         | 0.8066        | 0.7321       | 0.8131         | 0.7016        |
> > > > > | YaTC-LwF     | 0.8607        | 0.8443       | 0.8962         | 0.8248        |
> > > > > | **LENS(Ours)**   | **0.9548**    | **0.9243**   | **0.9296**     | **0.8914**    |
> > > > >
> > > > > **Table 11: Few-shot performance of 5 unseen classes on Task 6 (CrossPlatform-Android-APP Classification)**
> > > > > | Models       | 5-shot ACC | 5-shot F1 | 10-shot ACC | 10-shot F1 |
> > > > > | ------------ | ---------- | ---------- | ---------- | ---------- |
> > > > > | ET-BERT      | 0.8070      | 0.6978     | 0.7204     | 0.5633     |
> > > > > | ET-BERT-LwF  | 0.8682     | 0.6647     | 0.8350      | 0.7346     |
> > > > > | YaTC         | 0.6934     | 0.5753     | 0.6883     | 0.5333     |
> > > > > | YaTC-LwF     | 0.8042     | 0.7301     | 0.7398     | 0.6108     |
> > > > > | **LENS(Ours)**   | **0.9267** | **0.8259** | **0.9181** | **0.8234** |
> > > > >
> > > > > In both Table 10 and 11, Lens consistently and substantially outperforms all baselines across both datasets and both shot settings. On Task 8, Lens improves over the second-best baseline by **+9.41%** AC / **+8.00%** F1 (5-shot) and **+3.34%** AC / **+6.66%** F1 (10-shot). On Task 6, the gains are **+5.85%** AC / **+9.58%** F1 (5-shot) and **+8.31%** AC / **+8.88%** F1 (10-shot).
> > > > >
> > > > > Firstly, both ET-BERT and YaTC without LwF fail to balance old and new classes well. They achieve acceptable performance on new classes at the cost of degraded accuracy on old ones. Furthermore, their LwF variants partly address this via KL-divergence regularization and extend to new classes gradually, but still perform worse than our Lens due to the distribution shifts in the MLP-based classification heads as new classes are introduced. Secondly, Lens reframes the classification as a **closed-ended generation task**, learning label distributions at the decoder side instead of a fixed MLP head. This aligns directly with the extensibility motivation discussed in our response to Q4, which mitigates the learning-forgetting conflicts in MLP heads.

---

> > > > > > ### Author Response · Authors · 2026-04-23
> > > > > >
> > > > > > **Q6: The paper should analyze the trade-off between adapting to new classes and retaining performance on old classes. Although the paper compares against LwF-style baselines, it does not provide enough analysis of whether Lens itself suffers from forgetting or instability during incremental updates.**
> > > > > >
> > > > > > **A6:** We further conducted ablation experiments to show (i)Lens's trade-off between extensibility to new classes and maintaining old classes' performance in 3 and 5 unseen class scenarios, and (ii) whether Lens itself suffers from forgetting during incremental updates.
> > > > > >
> > > > > > To better quantify Lens's trade-off on both new and old classes, we use the **harmonic mean (HM)** of old class and new class performance, defined as HM = 2·(Old·New)/(Old+New). HM penalizes methods that sacrifice one side for the other.
> > > > > >
> > > > > > **(i) Trade-off between old and new classes**
> > > > > > We evaluate Lens under both 3 and 5 unseen classes scenarios on Task 8 (CrossPlatform-iOS Application Classification) and Task 6 (CrossPlatform-Android Application Classification) against two LwF baselines.
> > > > > >
> > > > > > **Table 12a: The performance of the 3-unseen classes on Task 8 (CrossPlatform-IOS-APP Classification)**
> > > > > >
> > > > > > | Method       | Old ACC | Old F1 | New ACC | New F1 | **HM-ACC** | **HM-F1** |
> > > > > > |--------------|:-------:|:------:|:-------:|:------:|:----------:|:---------:|
> > > > > > | ET-BERT-LwF  | **0.9600**  | **0.9254** | 0.6201  | 0.3720 | 0.7539     | 0.5308    |
> > > > > > | YaTC-LwF     | 0.9168  | 0.8612 | 0.8116  | 0.2840 | 0.8610     | 0.4274    |
> > > > > > | **LENS (Ours)**  | 0.8801  | 0.8819 | **1.0000**  | **0.7662** | **0.9362** | **0.8200** |
> > > > > >
> > > > > > **Table 12b: The performance of the 5-unseen class on Task 8 (CrossPlatform-IOS-APP Classification)**
> > > > > > | Method       | Old ACC | Old F1 | New ACC | New F1 | **HM-ACC** | **HM-F1** |
> > > > > > |--------------|:-------:|:------:|:-------:|:------:|:----------:|:---------:|
> > > > > > | ET-BERT-LwF  | **0.9258**  | **0.8662** | 0.6730  | 0.2685 | 0.7793     | 0.4099    |
> > > > > > | YaTC-LwF     | 0.8966  | 0.8360 | 0.9409  | 0.5889 | **0.9182**    | 0.6911    |
> > > > > > | **LENS (Ours)**  | 0.8446  | 0.8424 | **0.9916**  | **0.7764** | **0.9122** | **0.8080** |
> > > > > >
> > > > > > **Table 13a: The performance of the 3-unseen class on Task 6 (CrossPlatform-Android-APP Classification)**
> > > > > > | Method       | Old ACC | Old F1 | New ACC | New F1 | **HM-ACC** | **HM-F1** |
> > > > > > |--------------|:-------:|:------:|:-------:|:------:|:----------:|:---------:|
> > > > > > | ET-BERT-LwF  | 0.8889  | 0.7106 | 0.5954  | 0.2812 | 0.7131     | 0.4028    |
> > > > > > | YaTC-LwF     | 0.8674  | 0.6954 | 0.7264  | 0.1394 | 0.7907     | 0.2323    |
> > > > > > | LENS (Ours)  | **0.9479**  | **0.8648** | **0.9862**  | **0.9391** | **0.9667** | **0.9004** |
> > > > > >
> > > > > > **Table 13b: The performance of the 5-unseen class on Task 6 (CrossPlatform-Android-APP Classification)**
> > > > > > | Method       | Old ACC | Old F1 | New ACC | New F1 | **HM-ACC** | **HM-F1** |
> > > > > > |--------------|:-------:|:------:|:-------:|:------:|:----------:|:---------:|
> > > > > > | ET-BERT-LwF  | **0.9172**  | 0.7705 | 0.6065  | 0.3880 | 0.7302     | 0.5161    |
> > > > > > | YaTC-LwF     | 0.8054  | 0.6105 | 0.6389  | 0.2785 | 0.7126     | 0.3824    |
> > > > > > | LENS (Ours)  | 0.9098  | **0.8257** | **0.9768**  | **0.8541** | **0.9421** | **0.8397** |
> > > > > >
> > > > > > **Across all four settings, Lens achieves the best HM-F1 (ranging from 80.80% to 90.04%) and comparable best HM-ACC (ranging from 91.22% to 96.67%).** The LwF baselines preserve old-class performance at the cost of poor new class adaptation. For instance, ET-BERT-LwF's new-class F1 falls in the range of 0.2685 to 0.3880, and YaTC-LwF's new-class F1 reaches only 0.1394 on Task 6 (3-Unseen). However, Lens extends well to new classes, achieving new-class F1 up to 0.9391 on Task 6. At the same time, Lens retains competitive old-class accuracy and F1, yielding the most balanced performance between old and new classes across both tasks and scenarios. This is because we reframe the classification as a closed-ended generation task to mitigate the distribution shift from new classes.
> > > > > >
> > > > > > **(ii) Does Lens itself forget when extended to new classes?**
> > > > > > To directly show Lens's forgetting, we compare Lens's old-class performance when the number of newly added classes increases from 3 to 5 on the same task.
> > > > > >
> > > > > > **Table 14: Lens's old-class performance when extended with 3 vs. 5 new classes.**
> > > > > > | Task | Old ACC (3→5) | Δ ACC | Old F1 (3→5) | Δ F1 |
> > > > > > |---|:---:|:---:|:---:|:---:|
> > > > > > | Task 8 | 0.8801 → 0.8446 | −3.55% | 0.8819 → 0.8424 | −3.95% |
> > > > > > | Task 6 | 0.9479 → 0.9098 | −3.81% | 0.8648 → 0.8257 | −3.91% |
> > > > > >
> > > > > > As shown in Table 14, Lens's old-class performance drops by only **3–4%** on both ACC and F1, demonstrating that Lens is stable on old classes when extended with new classes, and does not exhibit catastrophic forgetting. We attribute this stability to our reframing of classification as a generation task at the decoder, which accommodates new classes while preserving the learned representations of old ones.

---

### Decision · Action_Editor_zhGn · 2026-06-21

**Recommendation:** Accept as is

**Additional Comments:**

The reviewers raised several useful concerns during the review process, including the strength of the knowledge-guided framing, the need for stronger baselines, the isolation of different architectural components, low-resource new-class adaptation, and the trade-off between new-class adaptation and old-class retention. The authors have addressed these points through additional experiments, ablation studies, and clarifications in the revised manuscript.

Overall, the revised paper is technically sound, empirically well supported, and relevant to TMLR. The contribution is primarily a solid domain-specific foundation model for network traffic classification and generation, with practical value and convincing experimental validation. I recommend acceptance.

**Audience:**

Yes

**Audience Explanation:**

The paper should be of interest to a subset of the TMLR audience, especially researchers working on domain-specific foundation models, sequence modeling, representation learning for structured data, cybersecurity applications, and generation-based classification. Although network traffic modeling is a specialized application domain, the paper presents a concrete example of injecting domain knowledge into pretraining through field-level masking and unifying classification and generation in an encoder-decoder framework.

**Claims And Evidence:**

Yes

**Claims Explanation:**

The claims are sufficiently supported by the empirical evidence and the revision has addressed the main reviewer concerns. The paper evaluates Lens on a broad set of network traffic classification and generation benchmarks, including 12 classification tasks and multiple generation tasks. The reported results show strong average classification performance and consistent improvements over relevant baselines, including ET-BERT, YaTC, LwF variants, and TrafficLLM in several settings.

During revision, the authors added or clarified several important pieces of evidence: ablations for knowledge-guided masked span prediction, contextual input, classification-as-generation, architecture choices against a decoder-only baseline, few-shot new-class extension experiments, and old/new class trade-off analysis. These additions make the evidence more convincing, especially for the claims about network-specific semantic modeling and extensibility to novel classes. Some limitations remain, such as scalability to very large label spaces and potential delayed patterns in long flows, but these limitations are now reasonably discussed and do not undermine the main conclusions.